# Fast Training of Sinusoidal Neural Fields via Scaling Initialization

**Taesun Yeom,**[*] **Sangyoon Lee**[*] **& Jaeho Lee**[†]
Pohang University of Science and Technology (POSTECH)
{tsyeom,sangyoon.lee,jaeho.lee}@postech.ac.kr

## Abstract

Neural fields are an emerging paradigm that represent data as continuous functions parameterized by neural networks. Despite many advantages, neural fields often have a high training cost, which prevents a broader adoption. In this paper, we focus on a popular family of neural fields, called sinusoidal neural fields (SNFs), and study how it should be initialized to maximize the training speed. We find that the standard initialization scheme for SNFs—designed based on the signal propagation principle—is suboptimal. In particular, we show that by simply multiplying each weight (except for the last layer) by a constant, we can accelerate SNF training by $10\times$. This method, coined *weight scaling*, consistently provides a significant speedup over various data domains, allowing the SNFs to train faster than more recently proposed architectures. To understand why the weight scaling works well, we conduct extensive theoretical and empirical analyses which reveal that the weight scaling not only resolves the spectral bias quite effectively but also enjoys a well-conditioned optimization trajectory.

## 1 Introduction

Neural field (NF) is a special family of neural networks designed to represent a single datum (Xie et al., 2022). Precisely, NFs parametrize each datum with the weights of a neural net which is trained to fit the mapping from spatiotemporal coordinates to corresponding signal values. Thanks to their versatility and low memory footprint, NFs have been rapidly adopted in various domains, including high-dimensional computer vision and graphics (Sitzmann et al., 2019; Mildenhall et al., 2020; Poole et al., 2023), physics-informed machine learning (Wang et al., 2021; Serrano et al., 2023), robotics (Simeonov et al., 2022), and non-Euclidean signal processing (Rebain et al., 2024).

For neural fields, the *training speed* is of vital importance. Representing each datum as an NF requires tedious training of a neural network, which can take up to several hours of GPU-based training (Mildenhall et al., 2020). This computational burden becomes a critical obstacle toward adoption to tasks that involve large-scale data, such as NF-based inference or generation (Ma et al., 2024; Papa et al., 2024). To address this issue, many research have been taken to accelerate NF training, including fast-trainable NF architectures (Sitzmann et al., 2020b; Müller et al., 2022), meta-learning (Sitzmann et al., 2020a; Chen & Wang, 2022), and data transformations (Seo et al., 2024).

Despite such efforts, a critical aspect of NF training remains understudied: How should we *initialize* NFs for the fastest training? While the importance of a good initialization has been much contemplated in classic deep learning literature (Glorot & Bengio, 2010; Zhang et al., 2019), such understandings do not immediately answer our question, for many reasons. First, conventional works mostly focus on how initialization affects the signal propagation properties of deep networks with hundreds of layers, while most NFs use shallow models with only a few layers (Mildenhall et al., 2020). Second, conventional works aim to maximize the model quality at convergence, while in NFs it is often more important how fast the network achieves certain fidelity criterion (Müller et al., 2022). Lastly, NFs put great emphasis on how the model performs on seen inputs (*i.e.,* training loss) and leaves how well it interpolates between coordinates (*i.e.,* generalizability) as a secondary issue, while conventional neural networks care exclusively on the test performance.

---

[*]Equal contribution.
[†]Corresponding author.

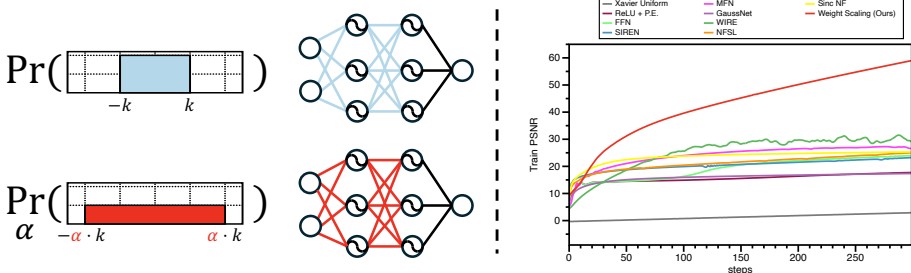

Figure 1. **A simple weight scaling accelerates training.** The proposed weight scaling scales up the initial weights of an SNF by the factor of $\alpha$, except for the last layer (left panel). The weight scaled SNF significantly speeds up training across a variety of methods (right panel: train PSNR curve for a single Kodak image).

To fill this gap, in this paper, we investigate how the initialization affects the training efficiency of neural fields. In particular, we focus on initializing *sinusoidal neural field* (SNF), *i.e.*, multilayer perceptrons with sinusoidal activation functions (Sitzmann et al., 2020b). SNF is one of the most widely used NFs, due to its versatility and parameter-efficiency; the architecture works as a strong baseline in a wide range of data modalities (Grattarola & Vandergheynst, 2022; Kim et al., 2024; Rußwurm et al., 2024) and is used as an indispensable building block of many state-of-the-art neural field frameworks (Guo et al., 2023; Schwarz et al., 2023; Pal et al., 2024).

**Contribution.** We first discover that the current practice of SNF initialization is highly suboptimal in terms of the training speed (Sitzmann et al., 2020b). Precisely, we find that one can accelerate the training speed of SNFs by up to $10\times$, simply by scaling up the initial weights of all layers by a well-chosen factor. This extremely simple tuning, coined *weight scaling* (WS), provides a much greater speedup than the conventional SNF tuning strategy, *i.e.*, scaling up the frequency multiplier of the activation functions (called $\omega_0$ in Sitzmann et al. (2020b)). Furthermore, this speedup can be achieved without any degradation in the generalization capabilities of the model, if we keep the scaling factor at a moderate level. Interestingly, experiments suggest that the optimal scaling factor, which strikes the balance between speedup and generalization, mainly depends on the physical scale of the NF workload (*e.g.*, data resolution and model size) and remains similar over different data.

To demystify why the weight scaling accelerates the SNF training, we conduct an in-depth analysis. In particular, we focus on understanding how the effects of WS on SNFs are similar yet different from ReLU nets, where the so-called *lazy training* phenomenon takes place (Chizat et al., 2019)— weight-scaled ReLU nets behave similarly to kernels, enjoying exponential convergence in theory for shallow nets but failing to achieve good accuracy at practical scales due to ill-conditioned trajectories. Our theoretical and empirical analyses highlight several key differences that sinusoidal activations bring, such as: (1) Weight-scaled SNFs also preserve activation distributions over layers, facilitating the gradient propagation when training a deep model (Proposition 1). (2) Weight scaling increases not only the frequency of each basis, but also increases the relative magnitude of higher-order harmonics, which helps fit the high-frequency components abundant in natural signals (Lemma 3 and Section 4.2). (3) Weight-scaled SNFs enjoy even better-conditioned trajectories than the unscaled models, as captured by the eigenspectrum analysis (Section 4.3).

In a sense, our study constitutes a call for rethinking the role of the initialization for neural fields, by demonstrating how dramatically suboptimal the current schemes can be in key performance metrics for NFs. We hope that our work will further inspire more sophisticated initialization algorithms.

## 2 RELATED WORK

**Training efficiency of neural fields.** There are two prevailing strategies to accelerate NF training. The first approach is to adopt advanced NF architectures (Fathony et al., 2021; Dou et al., 2023), which mitigate the spectral bias of neural nets that hinders fitting high-frequency signals (Rahaman et al., 2019; Basri et al., 2019; Xu et al., 2020; Cao et al., 2021; Choraria et al., 2022). Popular examples include utilizing specially designed positional encodings (Tancik et al., 2020; Müller et al., 2022) or activation functions (Sitzmann et al., 2020b; Ramasinghe & Lucey, 2022; Liu et al., 2024; Saratchandran et al., 2024c). Another strategy is to amortize the training cost via meta-learning, *e.g.,* by training a fast-trainable initialization (Sitzmann et al., 2020a; Tancik et al., 2021) or constructing

an auxiliary model that predicts NF parameters from the datum (Chen & Wang, 2022). This work aims to accelerate NF training through a better initialization. Unlike meta-learned initializations, however, our aim is to understand how the initialization (its scale, specifically) can facilitate training by analyzing how it affects optimization dynamics, without meta-learning involved.

**Neural network initialization.** Early works on network initialization focus on preserving signals through layers via principled weight scaling (Glorot & Bengio, 2010; He et al., 2015) or orthogonalization (Xiao et al., 2018). Recent studies explore its impact on the implicit bias of SGD (Vardi, 2023), where initialization scale determines whether training occurs in the *rich* or *kernel* regime (Jacot et al., 2018; Woodworth et al., 2020; Varre et al., 2023). While aligned with this line of research, we primarily investigate how initialization affects training speed, particularly in neural fields.

**Initializing sinusoidal neural fields.** There is a limited number of works that study the initialization for SNFs. Sitzmann et al. (2020b) proposes an initialization scheme for SNF, designed in the signal propagation perspective. Tancik et al. (2021) develops a meta-learning algorithm to learn initializations for SNFs. Lee et al. (2021) extends the algorithm for sparse networks. Most related to our work, Saratchandran et al. (2024b) proposes an initialization scheme that aims to facilitate the parameter-efficiency of SNFs. Different from this work, our work primarily focuses on the training efficiency, and achieves much faster training speed empirically.

**Sinusoidal activation functions in ML.** Beyond their well-known use in NF literature (Sitzmann et al., 2020b; Ashkenazi & Treister, 2024), sinusoidal activation functions have also been studied in other domains. Examples include Fourier neural networks (Gashler & Ashmore, 2014), Bayesian neural networks and out-of-distribution detection (Meronen et al., 2021), low-rank learning (Ji et al., 2025), neural compression (Thrash et al., 2025), and generative models (Chan et al., 2021).

## 3 NEURAL FIELD INITIALIZATION VIA WEIGHT SCALING

In this section, we formally describe the weight scaling for a sinusoidal neural field initialization, and briefly discuss the theoretical and empirical properties of the method.

### 3.1 NETWORK SETUP AND NOTATIONS

We begin by introducing the necessary notations and problem setup. A sinusoidal neural field is a multilayer perceptron (MLP) with sinusoidal activation functions (Sitzmann et al., 2020b). More concretely, suppose that we have an input $X \in \mathbb{R}^{d_0 \times N}$, where $N$ is the number of $d_0$-dimensional coordinates. Then, an $l$-layer SNF can be characterized recursively via:

$$f(X; \mathbf{W}) = W^{(l)} Z^{(l-1)} + b^{(l)}, \qquad Z^{(i)} = \sigma_i(W^{(i)} Z^{(i-1)} + b^{(i)}), \quad i \in [l-1], \tag{1}$$

parametrized by $\mathbf{W} = (W^{(1)}, b^{(1)}, \ldots, W^{(l)}, b^{(l)})$, where each $W^{(i)} \in \mathbb{R}^{d_i \times d_{i-1}}$ denotes the weight matrix, $b^{(i)} \in \mathbb{R}^{d_i}$ denotes the bias, and $Z^{(i)} \in \mathbb{R}^{d_i \times N}$ denotes the activation of the $i$th layer. Here, we let $Z^{(0)} = X$, and the activation function $\sigma_i : \mathbb{R}^{d_i} \to \mathbb{R}^{d_i}$ are the sinusoidal activations applied entrywise, with some frequency multiplier $\omega$. The standard practice is to use a different frequency multiplier for the first layer. Precisely, we apply the activation function $\sigma_1(x) = \sin(\omega_0 \cdot x)$ for the first layer, and $\sigma_i(x) = \sin(\omega_h \cdot x)$ for all other layers, *i.e.*, $i \neq 1$.

The SNF is trained using the gradient descent, with the mean-squared error (MSE) as a loss function. We assume that the input coordinates of the training data (*i.e.*, the column vectors of the input data) have been scaled to lie inside the hypercube $[-1, +1]^{d_0}$, as in Sitzmann et al. (2020b).

### 3.2 STANDARD INITIALIZATION

The standard initialization scheme for the sinusoidal neural fields independently draws each weight entry from a random distribution (Sitzmann et al., 2020b). Precisely, the weight matrices are initialized according to the scaled uniform distribution as

$$w_{j,k}^{(1)} \overset{\text{i.i.d.}}{\sim} \text{Unif}\left(-\frac{1}{d_0}, \frac{1}{d_0}\right), \qquad w_{j,k}^{(i)} \overset{\text{i.i.d.}}{\sim} \text{Unif}\left(-\frac{\sqrt{6}}{\omega_h \sqrt{d_{i-1}}}, \frac{\sqrt{6}}{\omega_h \sqrt{d_{i-1}}}\right), \tag{2}$$

where $w_{j,k}^{(i)}$ denotes the $(j, k)$-th entry of the $i$th layer weight matrix $W^{(i)}$.

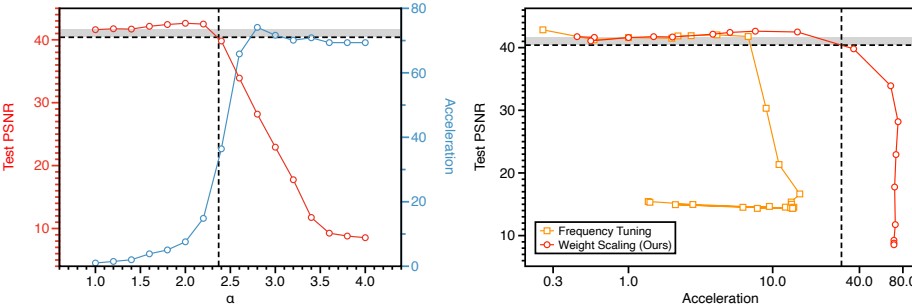

(a) Weight scaling with various scaling factors  (b) Tradeoff curve of WS vs. frequency tuning

Figure 2. **Scaling factors and the speed-generalization tradeoff.** (a) As we increase the scaling factor $\alpha$, the training speed ($\circ$) tends to become faster while the interpolation performance ($\circ$) gets lower. Notably, however, there exists some range of $\alpha$ where we enjoy acceleration with negligible degradation in test PSNR. (b) Comparing with the frequency tuning ($\square$), the weight scaling ($\circ$) achieves a better tradeoff. Further experimental details about the tradeoff curve is provided in Appendix F.1.

Importantly, the range of the uniform initialization (Eq. (2)) is determined based on the *distribution preserving* principle, which aims to make the distribution of the activation $Z_i$ similar throughout all layers. In particular, Sitzmann et al. (2020b, Theorem 1.8) formally proves that the choice of the numerator (*i.e.,* $\sqrt{6}$) ensures the activation distribution to be constantly $\arcsin(-1, 1)$ throughout all hidden layers of the initialized sinusoidal neural field.

**Frequency tuning.** A popular way to tune the initialization (Eq. (2)) for the given data is by modifying the frequency multiplier of the first layer $\omega_0$. This method, which we call *frequency tuning*, affects the frequency spectrum of the initial SNF, enhancing the fitting of signals within certain frequency range. It is not typical to tune $\omega_h$, as changing its value does not change the initial SNF; the effects are cancelled out by the scaling factors in the initialization scheme.

### 3.3 Initialization with weight scaling

In this paper, we develop a simple generalization of the standard initialization scheme. In particular, we consider *weight scaling*, which multiplies a certain scaling factor $\alpha \geq 1$ to the initialized weights in all layers except for the last layer. More concretely, the initialization scheme is

$$w_{j,k}^{(1)} \overset{\text{i.i.d.}}{\sim} \mathrm{Unif}\left(-\frac{\alpha}{d_0}, \frac{\alpha}{d_0}\right), \qquad w_{j,k}^{(i)} \overset{\text{i.i.d.}}{\sim} \mathrm{Unif}\left(-\frac{\alpha\sqrt{6}}{\omega_h\sqrt{d_{i-1}}}, \frac{\alpha\sqrt{6}}{\omega_h\sqrt{d_{i-1}}}\right), \tag{3}$$

where we let $\alpha = 1$ for the last layer; this design is due to the fact that any scaling in the last layer weight leads to a direct scaling of the initial function $f \mapsto \alpha f$, which is known to be harmful for the generalizability, according to the theoretical work of Chizat et al. (2019). Empirically, we observe that this weight scaling is much more effective in speeding up the training than a recently proposed variant which scales up only the last layer of the NF (Saratchandran et al., 2024b); see Section 5.

There are two notable characteristics of the weight-scaled initializations.

**(1) Distribution preservation.** An attractive property of the proposed weight scaling is that, for SNFs, any $\alpha \geq 1$ also preserves the activation distribution over the layers.[1] In particular, we provide the following result, which shows that the activation distributions will still be an $\arcsin(-1, 1)$.

**Proposition 1** (Informal; extension of Sitzmann et al. (2020b, Theorem 1.8)). *Consider an $l$-layer SNF initialized with the weight scaling (Eq. (3)). For any $\alpha \geq 1$, we have, in an approximate sense,*

$$Z^{(i)} \sim \arcsin(-1, 1), \qquad \forall i \in \{2, \ldots, l-1\}. \tag{4}$$

The detailed statement of this proposition and the numerical derivation will be given in Appendix A.

The proposition implies that we can tune the scaling factor $\alpha$ within the range $[1, \infty)$ without any concerns regarding preserving the forward propagated signal. Thus, we can safely adopt the weight scaling for training a very deep sinusoidal neural field.

---

[1]This point has also been noted in Sitzmann et al. (2020b), but has not been discussed quantitatively for $\alpha \geq 1$. We make this point concrete via numerical analyses on the large $\alpha$ case.

**(2) Speed vs. generalization.** Empirically, we observe that the weight scaling (Eq. (3)) allows us to explore for the "sweet spot" region, where we can enjoy the accelerated training, while not suffering from any degradation in the generalization performance on unseen coordinates.

The Fig. 2 illustrates this phenomenon for the case of image regression, where we train an SNF to fit a downsampled image until some desired peak signal-to-noise ratio (PSNR) is met, and then use the model to interpolate between seen pixel coordinates. We observe that by increasing $\alpha$ from 1, the loss on unseen coordinates increase, and the training speed increases as well. Notably, there exists some $\alpha > 1$ where the test PSNR remains similar while the training speed has strictly increased.

## 3.4    THE PARADIGM OF EFFICIENCY-ORIENTED INITIALIZATION

The latter phenomenon motivates us to develop an alternative perspective on a goodness of a neural field initialization, which may replace or be used jointly with the distribution preservation principle. More specifically, we consider an optimization problem

$$\max_{\alpha \geq 1} \text{speed}(\alpha)/\text{speed}(1) \quad \text{subject to} \quad \text{TestLoss}(\alpha) - \text{TestLoss}(1) \leq \varepsilon, \qquad (5)$$

where $\text{speed}(\alpha)$ denotes the training speed (*e.g.,* a reciprocal of the number of steps required) of SNF when initialized with $\alpha$ weight scaling, and $\text{TestLoss}(\alpha)$ denotes the loss on the unseen coordinates when trained from the initialization. Given this objective, the key questions we try to address are:

- Why does the weight scaling accelerate the training of SNFs?                        ▷ See Section 4
- Is weight scaling generally applicable for other data modalities?                    ▷ See Section 5.1
- How can we select a good $\alpha$, without expensive per-datum tuning?               ▷ See Section 5.2

## 4    UNDERSTANDING THE EFFECTS OF THE WEIGHT SCALING

We now take a closer look at the weight scaling to understand how it speeds up the training of sinusoidal neural fields. Before diving deeper into our analyses, we first discuss why a conventional understanding does not fully explain the phenomenon we observed for the case of SNF.

**Comparison with lazy training.** At the first glance, it is tempting to argue that this phenomenon is connected to the "lazy training" (Chizat et al., 2019; Woodworth et al., 2020): Scaling the initial weights ($\mathbf{W} \mapsto \alpha \mathbf{W}$) amplifies the initial model functional ($f \mapsto \alpha^l f$) (Taheri et al., 2021), which in turn leads to the model to be linearized; then, one can show that the SGD converges at an exponential rate, *i.e.,* the fast training, with the trained parameters being very close to the initial one. However, there is a significant gap between this theoretical understanding and our observations for the SNFs:

- The proposed weight scaling does not amplify the initial functional, as sinusoidal activations are not positively homogeneous (*i.e.,* $\sigma(ax) \neq a \cdot \sigma(x)$, even for positive $a, x$), unlike ReLU. In fact, WS keeps the last layer parameters unscaled, making it further from any functional amplification, and empirically works better than a direct amplification of functional (Saratchandran et al., 2024b).
- At a practical scale, a naïve scaling of initial weights for ReLU networks leads to both high training and test losses, due to a high condition number (Chizat et al., 2019, Appendix C). In contrast, the training loss of the WS remains very small even at the practical tasks.

In light of these differences, we provide alternative theoretical and empirical analyses that can help explain why the WS accelerates training, unlike in ReLU nets. In particular, we reveal that:

- WS on SNFs has two distinctive effects from ReLU nets, in terms of the initial functional and the layerwise learning rate; the former is the major contributor to the acceleration (Section 4.1).
- At initialization, WS increases both (1) the frequency of each basis, and (2) the relative power of the high-frequency bases (Section 4.2).
- WS also results in a better-conditioned optimization trajectory (Section 4.3).

## 4.1    TWO EFFECTS OF WEIGHT SCALING ON SINUSOIDAL NETWORKS

There are two major differences in how the weight scaling impacts SNF training compared to its effect on conventional neural networks with positive-homogeneous activation functions, *e.g.,* ReLU. (1) *Initial functional*: Larger weights lead to higher frequency (low-level) features in SNF, whereas

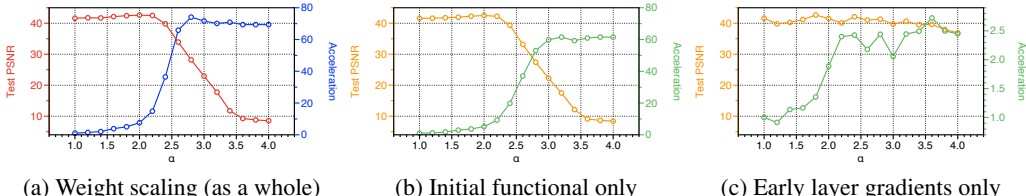

(a) Weight scaling (as a whole)     (b) Initial functional only     (c) Early layer gradients only

Figure 3. **Decoupling the effects of weight scaling on SNF.** We decouple the effect of weight scaling on how it amplifies the early layer gradients, from the effects of having a higher-frequency initial functional. We observe that the initial functional itself plays the key role, resulting in a much greater acceleration.

in ReLU nets scaling up does not change the neuronal directions (more discussions in Section 4.2). (2) *Larger early layer gradients*: Unlike ReLU nets, scaling up SNF weights has a disproportionate impact on the layerwise gradients; earlier layers receive larger gradients (see remarks below).

Motivated by this insight, we compare which of these two elements play a dominant role in speeding up the SNF training by decoupling these elements (Fig. 3). In particular, we analyze how changing only the initial weights—with the layerwise learning rates that are scaled down to match the rates of unscaled SNF—affects the speed-generalization tradeoff; we also compare with the version where the initial function stays the same as default, but the learning rates on early layers are amplified. From the results, we observe that the effect of having a good initial functional, with rich high-frequency functions, play a dominant role on speeding up the SNF training. The disproportionate learning rate also provides a mild speedup, while requiring less sacrifice in the generalizability.

**How are the early layer gradients amplified?** Essentially, this is also due to the lack of positive-homogeneity of the activation function. To see this, consider a toy two-layer neural net with width one $f(x) = w^{(2)}\sigma(w^{(1)}x)$. The loss gradient for each layer weight is proportional to the layerwise gradients of the function, which is $\nabla_{\mathbf{w}}f = \left(w^{(2)}x\sigma'(w^{(1)}x), \sigma(w^{(1)}x)\right)$. For ReLU activations, both entries of $\nabla_{\mathbf{w}}f$ grow linearly as we increase the scale of both weights. For $\sin$ activations, on the other hand, the first entry scales linearly, but the scale of the second entry remains the same. Following a similar line of reasoning, we can show that the first layer of a depth-$l$ sinusoidal network has a gradient that scales as $\alpha^l$, while the last layer gradient scales as $1$. We provide a more detailed discussion on the $\alpha$-dependency of layerwise gradients in Appendix B.3.

### 4.2 FREQUENCY SPECTRUM OF THE INITIAL FUNCTIONAL

We have seen that the initial functional has a profound impact on the training speed, but how does the weight scaling affect the functional? The answer is easy for two-layer nets; the weight scaling amplifies the frequency of each hidden layer neuron, enabling the model to cover wider frequency range with hardly updating the *features* generated by the first layer.

How about for deeper networks? Our theoretical and empirical analyses reveal that, for deeper SNFs, the weight scaling also increases the *relative power* of the higher-frequency bases. Thus, the weight-scaled networks may suffer less from the spectral bias, and can easily represent signals with large high-frequency components (Rahaman et al., 2019).

**Theoretical analysis.** We now show theoretically that the weight scaling increases the relative power of the higher-frequency bases. For simplicity, consider again a three-layer bias-free SNF

$$f(x; \mathbf{W}) = W^{(3)}\sin(W^{(2)}\sin(W^{(1)}x)), \tag{6}$$

with sinusoidal activation functions. Then, it can be shown that the overall frequency spectrum of the function $f(\cdot; \mathbf{W})$ can be re-expressed as the multiples of the first layer frequencies, via Fourier series expansion and the Jacobi-Anger identity (Yüce et al., 2022).

**Lemma 2** (Corollary of Yüce et al. (2022))**.** *The network Eq. (6) can be rewritten as*

$$f(x; \mathbf{W}) = 2W^{(3)}\sum_{\ell \in \mathbb{Z}_{odd}} J_{\ell}(W^{(2)})\sin(\ell W^{(1)}x), \tag{7}$$

*where $J_{\ell}(\cdot)$ denotes the Bessel function (of the first kind) with order $\ell$.*

Given the Bessel form (7), let us proceed to analyze the effect of weight scaling. By scaling the first layer weight, *i.e.*, $W^{(1)} \mapsto \alpha W^{(1)}$, we are increasing the frequencies of each sinusoidal basis by the factor of $\alpha$. That is, each frequency basis becomes a higher-frequency sinusoid.

Scaling the second layer weight, on the other hand, acts by affecting the magnitude of each sinusoidal basis, *i.e.*, $J_\ell(W^{(2)})$. For this quantity, we can show that the rate that the Bessel function coefficients evolve as we consider a higher order harmonics (*i.e.*, larger $\ell$) scales at the speed proportional to $\alpha^2$. That is, the relative power of the higher-order harmonics becomes much greater as we consider larger scaling factor $\alpha$. To formalize this point, let us further simplify the model (6) to have width one. Then, we can prove the following sandwich bound:

**Lemma 3** (Scaling of harmonics). *For any nonzero $W^{(2)} \in (-\pi/2, \pi/2)$, we have:*

$$\frac{(W^{(2)})^2}{(2\ell+2)(2\ell+4)} < \frac{J_{\ell+2}(W^{(2)})}{J_\ell(W^{(2)})} < \frac{(W^{(2)})^2}{(2\ell+1)(2\ell+3)}. \tag{8}$$

In other words, replacing $W^{(2)} \mapsto \alpha \cdot W^{(2)}$ increases the growth rate $J_{\ell+2}/J_\ell$ by $\alpha^2$. We note that there exists a restriction in the range $W^{(2)}$ which prevents us from considering very large $\alpha$. However, empirically, we observe that such amplification indeed takes place for the practical ranges of $\alpha$ where the generalization performance remains similar.

Summing up, the weight scaling has two effects on SNFs with more than two layers. First, the WS increases the frequencies of each sinusoidal basis by scaling the first layer weights. Second, the WS increases the relative weight of higher-frequency sinusoidal bases, by scaling the weight matrices of the intermediate layers.

**Empirical analysis.** To demonstrate that the increase of coefficients in the higher-order harmonics takes place indeed, we conduct a spectral analysis of an initialized SNF for natural image (Fig. 4). From the plot, we observe that the initialized SNFs with larger scaling factor $\alpha$ has a richer higher frequency spectrum. Moreover, the decay rate of the higher-frequency components, as we consider

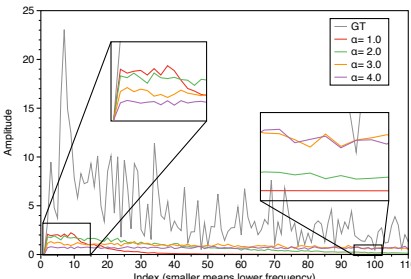

Figure 4. **Spectrum of initialized SNFs.** 1D-FFT of an initialized 5-layer SNF, with various levels of weight scaling factors.

increase the frequency level, is much smaller for larger $\alpha$; as a consequence, weight-scaled SNFs tend to have non-zero coefficients for very high frequency ranges, enabling a faster fitting to the natural audio signals (gray) with rich high-frequency components. Another

## 4.3 OPTIMIZATION TRAJECTORIES OF WEIGHT-SCALED NETWORKS

Now, we study the optimization properties of SNFs through the lens of the empirical neural tangent kernel (eNTK). We empirically analyze two properties of the eNTK eigenspectrum of the SNF, which are relevant to the training speed: (1) *condition number*, and (2) *kernel-task alignment*.

**Condition number.** Roughly, the condition number denotes the ratio $\lambda_{\max}/\lambda_{\min}$ of the largest and the smallest eigenvalues of the eNTK; this quantity determines the exponent of the exponential convergence rates of the kernelized models, with larger condition number meaning the slower training (Chizat et al., 2019). In Fig. 5a, we plot how the condition numbers of the eNTK evolve during the training for SNFs, under weight scaling or frequency tuning; for numerical stability, we take an average of top-5 and bottom-5 eigenvalues and take their ratio. We observe that the weight-scaled SNFs tend to have a much smaller condition number at initialization, which becomes even smaller during training. On the other hand, the SNF with the standard initialization suffers from a high condition number, which gets even higher during training. More discussion can be found in Appendix C.3.

We note that this behavior is in stark contrast with ReLU neural networks, where the weight scaling leads to a higher condition number, leading to a poor convergence (Chizat et al., 2019).

**Kernel-task alignment.** The kernel-task alignment—also known as the *energy concentration*—is the cumulative sum of the (normalized) dot products

$$\mathcal{E}(t) = \sum_{i:\lambda_i/\lambda_0 \geq t} (\phi_i^\top \mathbf{e})^2/\|\mathbf{e}\|_2^2 \tag{9}$$

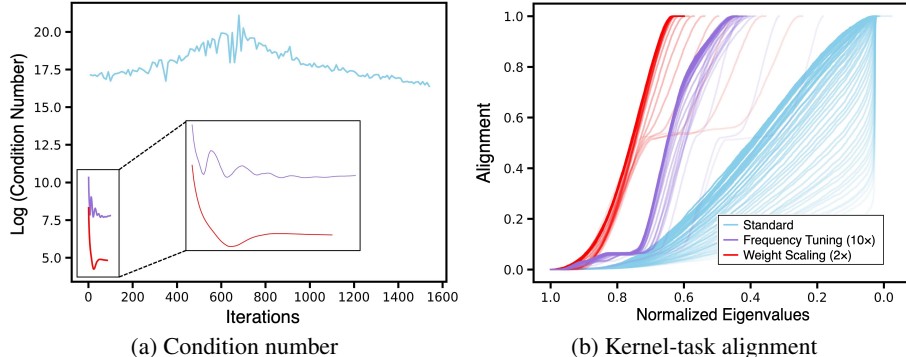

(a) Condition number          (b) Kernel-task alignment

Figure 5. **Eigenanalyses with eNTK.** (a) Weight scaling improves the conditioning of the SNF optimization at all SGD steps, greatly reducing the condition number. (b) Weight-scaled SNF enjoys a better kernel-task alignment throughout the training; darker lines indicate later iterations.

Table 1. **Weight scaling in various data domains.** We compare the training speed of the weight-scaled SNF against other baselines in various data domains. To evaluate the training speed, we train for a fixed number of steps and compare the training loss achieved. **Bold** denotes the best option, and underlined denotes the runner-up. We have experimented with five random seeds, and report the mean and the standard deviation.

| | | Image (PSNR) | | Occ. Field (IoU) | Spherical (PSNR) | Audio (PSNR) |
|---|---|---|---|---|---|---|
| | Activation | KODAK | DIV2K | Lucy | ERA5 | Bach |
| Xavier Uniform (Glorot & Bengio, 2010) | Sinusoidal | 0.46±0.10 | 0.39±0.10 | 0.0000±0.0000 | 4.11±0.66 | 7.77±0.20 |
| ReLU + P.E. (Mildenhall et al., 2020) | ReLU | 18.60±0.08 | 16.72±0.08 | 0.9896±0.0003 | 33.30±0.54 | 24.98±0.19 |
| FFN (Tancik et al., 2020) | ReLU | 20.52±0.60 | 19.81±0.48 | 0.9843±0.0020 | 38.69±0.27 | 16.66±0.28 |
| SIREN init. (Sitzmann et al., 2020b) | Sinusoidal | 24.58±0.05 | 22.86±0.06 | 0.9925±0.0001 | 38.72±0.07 | 37.37±3.11 |
| GaussNet (Ramasinghe & Lucey, 2022) | Gaussian | 21.94±2.48 | 19.22±0.14 | 0.9914±0.0005 | 38.56±0.51 | 27.47±2.10 |
| MFN (Fathony et al., 2021) | Wavelet | 28.54±0.12 | 26.42±0.10 | 0.9847±0.0003 | 36.89±0.80 | 16.16±0.05 |
| WIRE (Saragadam et al., 2023) | Wavelet | 28.94±0.21 | 28.20±0.13 | 0.9912±0.0005 | 31.27±0.53 | 16.83±1.85 |
| NFSL (Saratchandran et al., 2024b) | Sinusoidal | 24.93±0.07 | 23.39±0.09 | 0.9925±0.0001 | 38.92±0.07 | 37.17±2.88 |
| Sinc NF (Saratchandran et al., 2024c) | Sinc | 27.73±0.27 | 26.42±0.24 | 0.9936±0.0002 | 36.15±0.51 | 23.03±0.40 |
| Weight scaling (ours) | Sinusoidal | **42.83±0.35** | **42.03±0.41** | **0.9941±0.0002** | **45.28±0.03** | **45.04±3.23** |

where $\phi_i$ denotes the $i$-th eigenvector of the kernel and $\mathbf{e} = Y - f(X)$ denotes the vector of prediction errors in the pixels (Kopitkov & Indelman, 2020). This quantity measures how much of the residual signal is concentrated in the directions of large eigenvalues, which is theoretically easier to be expressed or optimized by kernel learning (Baratin et al., 2021; Yüce et al., 2022). In Fig. 5b, we compare the kernel-task alignments during the SNF training trajectories. We observe that the weight scaling gives a highly optimizable kernel throughout all stages of training.

## 5 EXPERIMENTS

In this section, we first address whether the weight scaling is effective in other data domains (Section 5.1); our answer is positive. Then, we discuss the factors that determine the optimal value of scaling factor $\alpha$ for the given target task (Section 5.2); we find that the optimal value does not depend much on the nature of each datum, but rather relies on the structural properties of the workload.

### 5.1 MAIN EXPERIMENTS

We validate the effectiveness of WS in different data domains by comparing against various neural fields. To compare the training speed, we compare the training accuracy for equivalent steps. In particular, we consider the following tasks and baselines. Other details can be found in Appendix F.4.

**Task: Image regression.** The network is trained to approximate the signal intensity $c$, for each given normalized 2D pixel coordinates $(x, y)$. For our experiments, we use Kodak (Kodak, 1999) and DIV2K (Agustsson & Timofte, 2017) datasets. Each image is resized to a resolution of $512 \times 512$ in grayscale, following Lindell et al. (2022); Seo et al. (2024). We report the training PSNR after a full-batch training for 150 iterations. For all NFs, we use five layers with width 512.

**Task: Occupancy field.** The network is trained to approximate the occupancy field of a 3D shape, *i.e.*, predict '1' for occupied coordinates and '0' for empty space. We use the voxel grid of size $512 \times$

$512 \times 512$ following Saragadam et al. (2023). For evaluation, we measure the training intersection-over-union (IoU) after 50 iterations on the 'Lucy' data from the 'Standard 3D Scanning Repository,' with the batch size 100k. We use NFs with five layers and width 256.

**Task: Spherical data.** We use 10 randomly selected samples from the ERA5 dataset, which contains temperature values corresponding to a grid of latitude $\phi$ and longitude $\theta$, using the geographic coordinate system (GCS). Following Dupont et al. (2022), the network inputs are lifted to 3D coordinates, *i.e.*, transforming $(\phi, \theta)$ to $(\cos(\phi) \cdot \cos(\theta), \cos(\phi) \cdot \sin(\theta), \sin(\phi))$. We report the training PSNR after 5k iterations of full-batch training. For NFs, we use five layers with width 256.

**Task: Audio data.** Audio is a 1D temporal signal, and the network is trained to approximate the amplitude of the audio at a given timestamp. We use the first 7 seconds of "Bach's Cello Suite No. 1," following Sitzmann et al. (2020b). We report the training PSNR after 1k iterations of full-batch training. For NFs, we use five layers with width 256.

**Baselines.** We compare the weight scaling against seven baselines; we have selected the versatile neural field methods, rather than highly specialized and heavy ones, such as Instant-NGP (Müller et al., 2022). More concretely, we use the following baselines: (1) *Xavier uniform*: the method by (Glorot & Bengio, 2010) applied on SNF, (2) *SIREN init.*: SNF using the standard initialization of SIREN (Sitzmann et al., 2020b), (3) *NFSL*: A recent initialization scheme motivated by scaling law (Saratchandran et al., 2024b). (4) *ReLU+P.E.*: ReLU nets with positional encodings (Mildenhall et al., 2020). (5) *FFN*: Fourier feature networks (Tancik et al., 2020). (6) *MFN*: multiplicative filter networks (Fathony et al., 2021). (7) *GaussNet*: MLP with Gaussian activations (Ramasinghe & Lucey, 2022). (8) *WIRE*: MLP with Gabor wavelet-based activations (Saragadam et al., 2023). (9) *Sinc NF*: MLP with sinc (*i.e.,* $\sin(\omega x)/\omega x$) activations (Saratchandran et al., 2024c). The baselines (1-3) use SNFs, while (4-9) uses other architectures with different activation functions.

**Result.** Table 1 reports the comprehensive results across various data domains. We observe that the weight scaling significantly outperforms all baselines throughout all tasks considered. In Appendix F.8, we provide further qualitative results on various modalities, where we find that the weight scaling enhances capturing the high frequency details. These results support our hypothesis that the richer frequency spectrum provided by the weight scaling mitigates the spectral bias (Section 4.2), thereby enabling a faster training of neural fields.

**Additional experiments.** We have also conducted experiments on the NF-based dataset construction tasks (Papa et al., 2024), novel view synthesis via neural radiance fields (NeRFs) (Mildenhall et al., 2020), and solving partial differential equations. We report the results and details for each experiment in Appendix F.5, Appendix F.6, and Appendix F.7 respectively.

## 5.2 ON SELECTING THE OPTIMAL SCALING FACTOR

Now, we consider the problem of selecting an appropriate scaling factor without having to carefully tune it for each given datum. In particular, we are interested in finding the right $\alpha$ that maximizes the training speed yet incurs only negligible degradations in generalizability. More concretely, consider the task of Kodak image regression and seek to select the $\alpha$ that approximately solves

$$\underset{\alpha}{\text{maximize}} \quad \text{speed}(\alpha) \qquad \text{subject to} \quad \text{TestPSNR}(\alpha) \geq 0.95 \cdot \text{TestPSNR}(1). \qquad (10)$$

where (again) the speed is the reciprocal of the number of steps required until the model meets the desired level of training loss, which we set to be the PSNR 50dB. Note that this optimization is similar to what has been discussed in Section 3.4.

In a nutshell, we observe in this subsection that this optimal scaling factor is largely driven by the structural properties of the optimization workload (such as model size and data resolution), and remains relatively constant over the data. This suggests that the strategy of tuning $\alpha$ on a small number of data and transferring it to other data in the same datasets can be quite effective.

**Structural properties.** In Fig. 6, we visualize how changing the data resolution ($512 \times 512$ by default), model width (512 by default), and model depth (4 by default) affects the value of optimal scaling factor. We observe that the higher data resolution and shallower model depth lead to a need for a higher scaling factor, with the width having very weak correlation. Commonly, we find that the resolution and depth are both intimately related to the frequency spectrum; higher resolution leads to a need for a higher frequency, and deeper networks can effectively utilize the relatively smaller

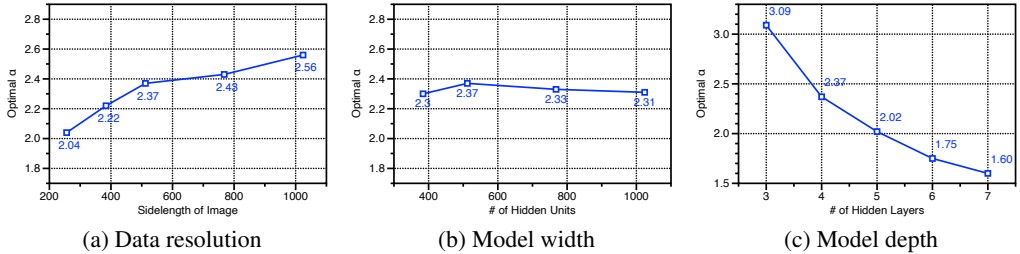

(a) Data resolution      (b) Model width      (c) Model depth

Figure 6. **Optimal scaling factor vs. structural properties of the optimization workload.** The data resolution, model width, and model depth have positive, no, and negative correlations with the optimal scaling factor.

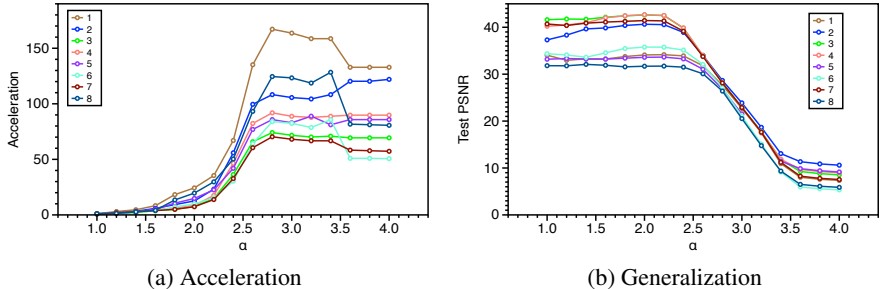

(a) Acceleration      (b) Generalization

Figure 7. **Scaling factor vs. the first 8 Kodak images.** The behavior of each datum with respect to the scaling factor $\alpha$ is quite similar, in both acceleration and generalization performances.

scaling factors to represent high-frequency signals. In a similar context, we provide an analysis of the impact of the learning rate in Appendix E.3.

**Data instances.** In Fig. 7, we observe that both curves are strikingly quite similar in shape, having saturation and inflection points at the similar scaling factor. This may be due to the fact that natural images tend to exhibit a similar frequency distribution.

## 6 CONCLUSION

In this paper, we revisit the traditional initialization scheme of sinusoidal neural fields, and propose a new method called weight scaling (WS), which scales the initialized weights by a specific factor. In practice, WS significantly accelerates the training speed of SNF across various data modalities, even outperforming recent neural field methods. We have discovered that the difference between the traditional and WS initialization comes from both the change in the frequency spectrum of the initial functional and the change in the layerwise learning rates. Through extensive theoretical and empirical analyses, we have revealed that the WS positively impacts the learning dynamics by maintaining a well-conditioned optimization path throughout the training process.

**Limitations and future directions.** The limitations of our work lie in the fact that the analysis has been conducted for only neural fields using sinusoidal activation functions, rather than general neural field architectures; how the WS affects on other architectures remains an open question (see Appendix E.2). Also, from a theoretical perspective, our analyses does not provide a definitive answer on the acceleration mechanisms, lacking an explicit convergence bound. As a final remark, our work is closely related to the *implicit bias of initialization scale* (Azulay et al., 2021), which is an emerging topic in understanding the neural network training. However, there remains a significant gap between theory and practice in this area, as the current interpretation is limited to shallow and homogeneous (or linear) networks (Varre et al., 2023; Kunin et al., 2024). The next step is to understand the implicit bias in networks with general activation functions, where the scale of initialization is not necessarily equivalent to output scaling. Another research direction is to understand the effect of weight scaling in general neural field architectures (Appendix E.2), which can sometimes improve performance or have the opposite effect. We leave these for future work.

## ACKNOWLEDGMENTS

This work was supported in part by the National Research Foundation of Korea (NRF) grant funded by the Korea government (MSIT) (No. RS-2023-00213710), in part by the Institute of Information & communications Technology Planning & Evaluation (IITP) grant funded by the Korea government (MSIT) (2022-0-00713), and in part by the NAVER-Intel Co-Lab; the work was conducted by POSTECH and reviewed by both NAVER and Intel.

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

# Appendix

## A    DETAILED STATEMENT AND THE DERIVATION OF PROPOSITION 1

In this section, we provide a more detailed statement and derivation of Proposition 1, where we have extended the distribution-preserving claims of Sitzmann et al. (2020b) on $\alpha = 1$ to the cases of $\alpha \geq 1$, thus covering the initialization with weight scaling.

We note that we generally follow the approach of Sitzmann et al. (2020b) in terms of the formalisms; we provide rough approximation guarantees (Lemmas 4 and 5), one of which relies on a numerical analysis of a function that is difficult to be expressed in a closed form (Lemma 5).

In particular, we are concerned with showing the following two points:

- **Activations to pre-activations.** We show that, for $\alpha \geq 1$, the pre-activation of a layer with weight-scaled uniform activations and $\arcsin$ input distributions are approximately distributed as a Gaussian distribution with variance scaled by $\alpha$ (Lemma 4).
- **Pre-activations to activations.** Then, we show that, for $\alpha \geq 1$, post-activation distribution of the Gaussian distribution with $\alpha$-scaled variance is distributed as an $\arcsin$ distribution (Lemma 5).

More formally, our lemma are as follows.

**Lemma 4** (Arcsin to Gaussian distributions). *Let the weights $W_l$ of the $l$-th layer be sampled i.i.d. from $\mathrm{Unif}(-c/\sqrt{n}, c/\sqrt{n})$, and the activation of the $(l-1)$-th layer $Z_{l-1}$ follow the i.i.d. $\arcsin(-1, 1)$. Then, the distribution of the pre-activation of the $l$-th layer $Y_l$ converges in distribution to a Gaussian distribution $\mathcal{N}(0, c^2/6)$ as the number of hidden units $n$ tends to infinity.*

**Lemma 5** (Gaussian to arcsin distributions, with numerical proof). *Let the pre-activation distribution $X$ has a Gaussian distribution $\mathcal{N}(0, \alpha^2)$ for some $\alpha \geq 1$. Then the activation distribution $Y = \sin(X)$ is distributed as $\arcsin(-1, 1)$.*

Note that, by plugging in $c = \alpha\sqrt{6}$ to the Lemma 4 as in the proposed weight scaling, we get the normal distribution $\mathcal{N}(0, \alpha^2)$ as an output distribution.

We provide the proofs of these lemma in Appendices A.1 and A.2, respectively. For all derivations, we simply ignore the issue of the frequency scaling term $\omega_0$, and simply let it equal to 1.

### A.1    PROOF OF LEMMA 4

For simplicity, let us denote a pre-activation of a neuron in the $l$th layer by $y$. Let the weight vector connected to this output neuron as $\mathbf{w} \in \mathbb{R}^n$, with each entries drawn independently from $\mathrm{Unif}(-c/\sqrt{n}, c/\sqrt{n})$. Also, let the corresponding input activation from the $(l-1)$th layer be denoted by $\mathbf{z} \in \mathbb{R}^n$, with each entry drawn independently from $\arcsin(-1, 1)$. Now, it suffices to show that $y = \mathbf{w}^\top \mathbf{z}$ converges in distribution to $\mathcal{N}(0, c^2/6)$ as $n$ goes to infinity.

To show this point, we simply invoke the central limit theorem for the i.i.d. case. For CLT to hold, we only need to check that each summand $(w_i z_i)$ of the dot product $\mathbf{w}^\top \mathbf{z} = \sum w_i z_i$ has a finite mean and variance.

$$\mathbf{w}^\top \mathbf{z} = \sum_{i=1}^{d} w_i z_i \tag{11}$$

has a finite mean and variance $\mathbf{w}^\top \mathbf{z}$ has a zero mean and a finite variance. We know that $\mathbb{E}[w_i z_i] = 0$, as two random variables are independent and $\mathbb{E}[w_i] = 0$. To show that the variance is finite, we can proceed as

$$\mathrm{Var}(w_i z_i) = \mathrm{Var}(w_i)\mathrm{Var}(z_i) = \frac{c^2}{3n}\frac{1}{2} = \frac{c^2}{6n}, \tag{12}$$

where the first equality holds as each variables are independent and zero-mean. Summing over $n$ indices, we get what we want.

### A.2    PROOF OF LEMMA 5

By the $3\sigma$ rule, 99.7% of the mass of the Gaussian distribution $\mathcal{N}(0, \alpha^2)$ lies within the finite support $[-3\alpha, 3\alpha]$. The CDF of $Y$ can be written as

$$F_Y(y) = P(Y \leq y) = P(\sin(X) \leq y). \tag{13}$$

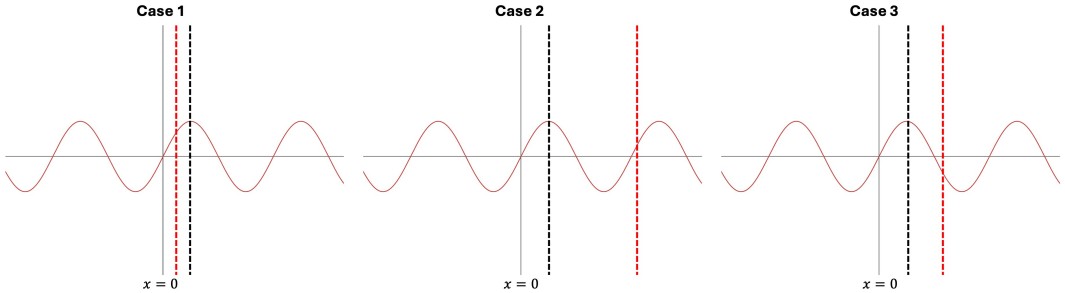

Figure 8. Visualization of the cases in Lemma 5. In $y = \sin(x)$, **Red line** : $x = 3\alpha$ and **Black line** : $x = \frac{\pi}{2}$

Different from the settings of Sitzmann et al. (2020b), the sinusoidal activation is not a bijection in general within the support $[-3\alpha, 3\alpha]$, *i.e.*, not directly invertible. Thus, we break it down into three cases (Fig. 8): Case 1, where the endpoint $3\alpha$ is less than $\pi/2$, Case 2, where $3\alpha$ is in the increasing region of $\sin(\cdot)$, and Case 3, where $3\alpha$ is in the decreasing region. We study each case:

Case 1 ($\alpha < \pi/6$). Here, we have

$$F_Y(y) = P(X \leq \arcsin y) = F_X(\arcsin y) \approx \frac{1}{2} + \frac{1}{2}\tanh\left(\frac{\beta}{\alpha}\arcsin y\right), \qquad (14)$$

where the last approximation invokes the Gaussian CDF approximation (Bowling et al., 2009). In this case, the CDF of $Y$ does not approximate the CDF of $\arcsin(-1, 1)$. That is, the second-order derivative of $F$ decreases in $[-1, 0]$ and increases in $[0, 1]$, while the CDF of the $\arcsin$ distribution behaves in an opposite way.

Now, suppose that $\alpha$ is sufficiently large ($\alpha \geq \pi/6$), *i.e.*, case 2 or case 3. Here, we will consider the nearest peak/valley that falls outside the $[-3\alpha, 3\alpha]$ interval. More concretely, let $\gamma$ be defined as the period index of such peak/valley, i.e.,

$$\gamma = \min\{\gamma_1, \gamma_2\}, \qquad \text{where} \qquad (15)$$

$$\gamma_1 = \min\left\{n \in \mathbb{N} \mid \frac{(4n+1)\pi}{2} \geq 3\alpha \ \cap \ \sin\left(\frac{(4n+1)\pi}{2}\right) = +1\right\} \qquad \text{(peak; case 2)} \quad (16)$$

$$\gamma_2 = \min\left\{n \in \mathbb{N} \mid \frac{(4n-1)\pi}{2} \geq 3\alpha \ \cap \ \sin\left(\frac{(4n-1)\pi}{2}\right) = -1\right\} \qquad \text{(valley; case 3)} \quad (17)$$

By considering the range up until such peaks/valleys, we can ensure that the probability of $X$ lying within this range is greater than 0.997.

Now, let's proceed to analyze the case 2; the case 3 can be handled similarly.

Case 2. We break down the CDF of $Y$ as

$$F_Y(y) = F_{Y,\text{center}}(y) + F_{Y,\text{inc}}(y) + F_{Y,\text{dec}}(y). \qquad (18)$$

Here, $F_{Y,\text{center}}(y)$ denotes the CDF of the bijective region, *i.e.*, $x \in [-\pi/2, \pi/2]$, and $F_{Y,\text{inc}}(y)$ denotes the region where the $\sin(\cdot)$ is increasing and $F_{Y,\text{dec}}(y)$ denotes where $\sin(\cdot)$ is decreasing. Then, each of these components can be written as[2]:

$$F_{Y,\text{center}}(y) = P(-\pi/2 \leq X \leq \arcsin y) = F_X(\arcsin y) - F_X(-\pi/2) \qquad (19)$$

$$F_{Y,\text{inc}}(y) = \sum_{n=1}^{\gamma_1}\Big[F_X(\arcsin y + 4n\pi/2) - F_X((4n-1)\pi/2)$$

$$+ F_X(\arcsin y - 4n\pi/2) - F_X(-(4n+1)\pi/2)\Big] \qquad (20)$$

$$F_{Y,\text{dec}}(y) = \sum_{n=1}^{\gamma_1}\Big[F_X((4n-1)\pi/2) - F_X(-\arcsin y + (4n-2)\pi/2)$$

$$+ F_X(-(4n-3)\pi/2) - F_X(-\arcsin y - (4n-2)\pi/2)\Big] \qquad (21)$$

---

[2]For notational brevity, we set $\alpha = 1$ just for a moment. It will appear again in Eq. (23).

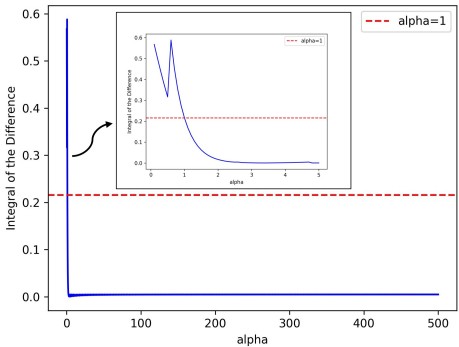

Figure 9. **Approximated CDF with increasing in $\alpha$.** Red line indicates the reference error. When $\alpha$ goes larger, the error converges to limit near zero.

Combining these terms, we get:

$$
\begin{aligned}
F_Y(y) = {} & F_X\left(\arcsin y\right) - F_X\left(-\pi/2\right) \\
& + \sum_{n=1}^{\gamma_1} \Big[ F_X(-(4n-3)\pi/2) - F_X(-(4n+1)\pi/2) \\
& + F_X(\arcsin y + 4n\pi/2) + F_X(\arcsin y - 4n\pi/2) \\
& - F_X(-\arcsin y + (4n-2)\pi/2) - F_X(-\arcsin y - (4n-2)\pi/2) \Big]
\end{aligned}
\tag{22}
$$

Using the logistic approximation and Taylor's first-order expansion at $z = \arcsin y = 0$, *i.e.,* $f(z) \approx \frac{f'(z)}{1!}(z - 0) + f(0)$, this can be further approximated by

$$
\begin{aligned}
F_Y(y) = {} & \left[ \frac{\beta}{2\alpha} h'(0) + \sum_{n=1}^{\gamma_1} \left( \frac{\beta}{\alpha} \left( h'\left( \frac{4n\beta\pi}{2\alpha} \right) + h'\left( \frac{(4n-2)\beta\pi}{2\alpha} \right) \right) \right) \right] z \\
& - \frac{1}{2} h\left( -\frac{\beta\pi}{2\alpha} \right) + \sum_{n=1}^{\gamma_1} \left[ \frac{1}{2} h\left( -\frac{(4n-3)\beta\pi}{2\alpha} \right) - \frac{1}{2} h\left( -\frac{(4n+1)\beta\pi}{2\alpha} \right) \right],
\end{aligned}
\tag{23}
$$

where $h$ is the $\tanh$ function. The above equation has variables $\gamma_1$ and $\alpha$. Thus, we can rewrite it as:

$$
F_Y(y) = f(\alpha) \cdot \arcsin y + g(\alpha),
\tag{24}
$$

$$
f(\alpha) = \frac{\beta}{2\alpha} h'(0) + \frac{\beta}{\alpha} \sum_{n=1}^{\gamma_1} \left[ h'\left( \frac{4n\beta\pi}{2\alpha} \right) + h'\left( \frac{(4n-2)\beta\pi}{2\alpha} \right) \right],
\tag{25}
$$

$$
g(\alpha) = -\frac{1}{2} h\left( -\frac{\beta\pi}{2\alpha} \right) + \frac{1}{2} \sum_{n=1}^{\gamma_1} \left[ h\left( -\frac{(4n-3)\beta\pi}{2\alpha} \right) - h\left( -\frac{(4n+1)\beta\pi}{2\alpha} \right) \right].
\tag{26}
$$

Now, we compare the $F_X(x)$ to ground truth CDF of $\arcsin(-1, 1)$, *i.e.,* $\frac{2}{\pi} \arcsin \sqrt{\frac{x+1}{2}}$. As the closed form solution is not obtainable, we numerically compute the difference in integration over input domain:

$$
\hat{\alpha} \in \left\{ \alpha \;\middle|\; \left| \int_{-1}^{1} \left| F_X(x, \alpha) - \frac{2}{\pi} \arcsin\left( \sqrt{\frac{x+1}{2}} \right) \right| dx \right. \right.
\tag{27}
$$

$$
\left. \left. \leq \int_{-1}^{1} \left| F_X(x, 1) - \frac{2}{\pi} \arcsin\left( \sqrt{\frac{x+1}{2}} \right) \right| dx \right\}.
\tag{28}
$$

This indicates that the RHS term represents the cumulative error from the default initialization, which is assumed to be valid for preserving the forward signal. If the LHS is smaller than the RHS for some value of $\hat{\alpha}$ that satisfies the inequality, it intuitively follows that the elements in $\hat{\alpha}$ provide a valid scale. In Fig. 9, we empirically demonstrate that as the scaling factor $\alpha$ increases, the error between the ground truth and the approximated CDF decreases with some limit near zero, which implies $\hat{\alpha} \in [1, \infty)$.

# B  PROOFS IN SECTION 4

## B.1  PROOF OF LEMMA 2

To formally prove this lemma, we first bring the generalized form from Yüce et al. (2022).

**Lemma 6** (From Yüce et al. (2022, Appendix A.2))**.** *Suppose that the model can be written as* $f(x) = W^{(3)} \sin\left(W^{(2)} \sin(W^{(1)}x)\right)$, *where* $W^{(1)} \in \mathbb{R}^{d_1}$, $W^{(2)} \in \mathbb{R}^{d_2 \times d_1}$, *and* $W^{(3)} \in \mathbb{R}^{1 \times d_2}$. *Then, the model can be equivalently written as:*

$$f(x) = \sum_{m=0}^{d_2-1} \sum_{i_1,\cdots,i_{d_1}=-\infty}^{\infty} \left(\prod_{k=0}^{d_1-1} J_k\left(w_k^{(2)}\right)\right) w_m^{(3)} \sin\left(\left(\sum_{k=0}^{d_1-1} i_k \cdot w_k^{(1)}\right) x\right), \qquad (29)$$

*where the lowercase* $w_k^{(i)}$ *denotes the k-th column of the matrix* $W^{(i)}$.

We specialize this lemma to the case of width-one SNF, *i.e.*, $d_1 = d_2 = 1$. We proceed as

$$f(x) = W^{(3)} \sin(W^{(2)} \sin(W^{(1)}x)) \qquad (30)$$

$$= 2W^{(3)} \sum_{k=0}^{\infty} J_{2k+1}(W^{(2)}) \sin((2k+1)W^{(1)}x), \qquad (31)$$

where the second equality is due to the Jacobi-Anger identity. By restricting the sum to $l \in \mathbb{Z}_{\text{odd}}$, we can further simplify as

$$f(x; \mathbf{W}) = 2W^{(3)} \sum_{\ell \in \mathbb{Z}_{\text{odd}}} J_\ell(W^{(2)}) \sin(\ell W^{(1)}x). \qquad (32)$$

## B.2  PROOF OF LEMMA 3

For notational simplicity, we denote $x := W^{(2)}$.

First, we note that the first root of the $\ell$th order Bessel function, denoted by $j_{\ell,1}$, satisfies the inequality $j_{\ell,1} > \pi/2$ for all $\ell \in \mathbb{N}$. Next, recall the results of Ifantis & Siafarikas (1990) that

$$\frac{x}{2(\ell+1)} < \frac{J_{\ell+1}(x)}{J_\ell(x)} < \frac{x}{2\ell+1}, \qquad \forall x \in (0, \pi/2). \qquad (33)$$

Similarly, for $J_{\ell+2}(x)$, we have

$$\frac{x}{2(\ell+2)} < \frac{J_{\ell+2}(x)}{J_{\ell+1}(x)} < \frac{x}{2\ell+3}, \qquad \forall x \in (0, \pi/2). \qquad (34)$$

Multiplying two inequalities, we get the desired claim.

## B.3  DETAILS OF SECTION 4.1

In this subsection, we provide more precise details of gradient bound of SNFs, with simple L-layer, 1-width network. Note that the bound can be generalized to arbitrary width, with matrix product computation.

Let $L$-layer simple SNF, $f(x)$, as in the main paper:

$$f(x) = W^{(L)} \sin(W^{(L-1)} \sin(\cdots \sin(W^{(1)}x)\cdots)). \qquad (35)$$

The network is trained with MSE loss function with gradient-based method, then the gradient of weight of each layer is computed as:

$$\frac{\partial L}{\partial W^{(k)}} = \frac{\partial L}{\partial f} \cdot \frac{\partial f}{\partial W^{(k)}} = R_0 \cdot \frac{\partial f}{\partial W^{(k)}}, \quad \text{for} \quad k \in [1, L], \qquad (36)$$

where $R_0$ denotes the initial residual, *i.e.*, $f(x) - Y_{\text{true}}$. We additionally assume that $R_0$ is independent from various initializations techniques. Then, $\partial f/\partial W^{(k)}$ only effects to gradients. By

the simple property of sine function: $\max_x \sin(x) = \max_x (d\sin(x)/dx) = 1$, the gradients are bounded with:

$$\left| \frac{\partial L}{\partial W^{(k)}} \right| = R_{t=0} \prod_{m=1}^{L-k} \left( \mathbb{1}_{L>k} W^{(L-m+1)} \cdot G_k(x) \cdot x^{L-k-1} + \mathbb{1}_{L=k} G_k(x) \right) \tag{37}$$

$$\leq R_{t=0} \prod_{m=1}^{L-k} \left( \mathbb{1}_{L>k} W^{(L-m+1)} \cdot \max_x G_k(x) \cdot x^{L-k-1} + \mathbb{1}_{L=k} \max_x G_k(x) \right) \tag{38}$$

$$= R_{t=0} \prod_{m=1}^{L-k} \left( \mathbb{1}_{L>k} W^{(L-m+1)} \cdot x^{L-k-1} + \mathbb{1}_{L=k} \right) \tag{39}$$

$$\leq R_{t=0} \prod_{m=1}^{L-k} \left( \mathbb{1}_{L>k} W^{(L-m+1)} + \mathbb{1}_{L=k} \right), \tag{40}$$

where $G_k(x)$ denotes a composite sinusoidal function from the gradient derivation process and $\mathbb{1}$ is a indicator function. Since the maximum value of a composite sinusoidal function is always equal to or less than 1, $|G_k(x)|$ is bounded by 1. Note that Eq. (40) is satisfied due to the bounds on input domain, i.e., $x \in [-1, 1]$.

## C   DETAILS OF NTK ANALYSIS AND BEYOND

### C.1   PRELIMINARIES

The Neural Tangent Kernel (NTK) framework views neural networks as kernel machines, where the kernel is defined as the dot-product of the gradients from two different data point (Jacot et al., 2018). In other words, each element of NTK can be expressed as $K(x_i, x_j) = \langle \nabla_{\mathbf{W}} f(x_i), \nabla_{\mathbf{W}} f(x_j) \rangle$. Assuming an infinitely wide network trained with an infinitesimally small learning rate (*i.e.,* gradient flow) and using a specific parameterization (*i.e.,* NTK parameterization), the NTK converges to a deterministic kernel that does not evolve during training. However, in practice, neural networks are composed of finite-width layers, which introduces challenges for analysis using NTK-style theory. In this context, we study the empirical Neural Tangent Kernel (eNTK), where we do not require the assumption of a stationary kernel. Recent researches (Kopitkov & Indelman, 2020; Ortiz-Jiménez et al., 2021; Baratin et al., 2021) show that, with eNTKs non-linearly evolve during training, this kernel can serve as a measure of the difficulty of learning a particular task.

### C.2   KERNEL REGRESSION WITH GRADIENT FLOW

First, we provide a well-founded spectral analysis of NTK regression. Assuming the network is trained with a constant training set $X$ and their label set $Y$ (*i.e.,* $((x_1, y_1), \cdots, (x_N, y_N))$, $x, y \in \mathbb{R}$), where $f(X; W_t) = f(W_t)$, and the loss function $L$ is defined as MSE. We can then express the time-dependent dynamics of the function using the NTK. To begin, we start with the equation for vanilla GD:

$$W_{t+1} = W_t - \eta \nabla_W L(W_k) \tag{41}$$

$$\frac{W_{t+1} - W_t}{\eta} = -\nabla_W L(W_k). \tag{42}$$

With infinitesimally small learning rate, *i.e.,* gradient flow, then:

$$\frac{dW_t}{dt} = -\nabla_W L(W_t)$$
$$= -\nabla_W f(W_t) \cdot (f(W_t) - Y_{\text{true}}). \tag{43}$$

Eq. 43 indicates the dynamics of the weights. Using a simple chain rule, we can modify the equation to represent the dynamics of the function, with the definition of NTK (i.e., $K_t \in \mathbb{R}^{N \times N}$):

$$\frac{df(W_t)}{dt} = \frac{df(W_t)}{dW_t} \cdot \frac{dW_t}{dt} = -\langle \nabla_W f(W_t), \nabla_W f(W_t) \rangle (f(W_t) - Y_{\text{true}})$$
$$= -K_t \cdot (f(W_t) - Y_{\text{true}}). \tag{44}$$

Note that the element-wise representation of NTK is can be written as $K_t(x_i, x_j) = \langle \nabla_W f(x_i, W_t), \nabla_W f(x_j, W_t) \rangle \in \mathbb{R}$, where $x_k$ denotes $k$-th data point in the entire training set $X$. Now, let $g(W_t) := f(W_t) - Y_{\text{true}}$, then Eq. 44 can be rewritten as:

$$\frac{dg(W_t)}{dt} = -K_t \cdot g(W_t). \tag{45}$$

Eq. 45 is a simple ordinary differential equation (ODE) and can be solved in the form of exponential growth by assuming the NTK is constant during update; $K_t = K_0$:

$$g(W_t) = e^{-t \cdot K_t} \cdot g(W_0) \tag{46}$$

$$f(W_t) - Y_{\text{true}} = e^{-t \cdot K_t} \cdot (f(W_0) - Y_{\text{true}}) \tag{47}$$

$$f(W_t) - Y_{\text{true}} = Q e^{-\Lambda t} Q^T \cdot (f(W_0) - Y_{\text{true}}) \tag{48}$$

$$Q^T (f(W_t) - Y_{\text{true}}) = e^{-\Lambda t} Q^T \cdot (f(W_0) - Y_{\text{true}}), \tag{49}$$

where $Q$ is matrix of concatenation of eigenfunctions, which behaves a projecting operator, and $\Lambda$ is diagonal matrix contains non-negative eigenvalues. Additionally, equality in Eq. 48 is due to the property of spectral decomposition, *i.e.,* $e^{K_t} = Q e^{\Lambda} Q^T$.

Eigenvectors compose orthonormal set in $\mathbb{R}^N$, and each residual components projected in each eigendirection. This directly indicates the convergence speed in each direction in $\mathbb{R}^N$. For example, we can explictly compute the remained error in direction $q_i$, denoted as $v_i$, at timestep $t$:

$$v_{t,i} = e^{-\lambda_i t} \cdot v_{0,i} \tag{50}$$

- We can observe that once the eigenspace is defined (at the initial state), the eigenvalue governs the training speed in each direction of the basis. Specifically, after an arbitrary time step $t$, the projected error in the direction corresponding to large eigenvalues decreases exponentially faster than in the case of small eigenvalues.

- Another important aspect is understanding the characteristics of each direction. Empirical and theoretical evidence has shown that the NTK eigenfunctions corresponding to large eigenvalues are low-frequency functions. As a result, neural network training is biased towards reducing error in low-frequency regions (Basri et al., 2019; 2020; Wang et al., 2021), providing strong evidence of *spectral bias* (Rahaman et al., 2019).

- A natural question arises, "what happens if the eigenvalue is large but the dot product between the corresponding eigenfunction and the residual is small?" This indicates that, despite the large eigenvalue, it contributes less to the reduction of the overall loss. This directly implies that we should consider not only the absolute scale of the eigenvalues but also the dot product (or alignment) between the eigenfunctions and the residual. In summary, we can conclude: *if there is high alignment in directions with large eigenvalues, training will be faster*.

Based on this analysis, let the timestep $\hat{t}$ that we want to evaluate the loss, then:

$$\hat{Q}^T(f(W_{t-\hat{t}}) - Y_{\text{true}}) = e^{-\hat{\Lambda}(t-\hat{t})}\hat{Q}^T(f(W_{\hat{t}}) - Y_{\text{true}}). \tag{51}$$

We evaluate the eNTK at every discrete timestep until the training ends. Specifically, we approximate the flow of error over the interval $t - \hat{t} \in [0, 1]$. This setting allows the kernel to evolve during training and enables us to further explore the evolution of its eigenspace. For computation of NTKs in our work, we use 'Neural Tangent Library' (Novak et al., 2020).

## C.3 Insights into eNTK eigenvalue

In this subsection, we focus on the eNTK of two-layer sinusoidal networks with the scaling factor $\alpha$, *i.e.,* $f(x; \mathbf{W}) = W_2 \sin(\alpha W_1 x)$. The resulting kernel is a sum of two dot-product kernels

$$K_{\text{NTK}}(x, x') = \underbrace{\langle \sin(\alpha W_1 x), \sin(\alpha W_1 x') \rangle}_{K} + \underbrace{\langle \alpha W_2 x \cos(\alpha W_1 x), \alpha W_2 x \cos(\alpha W_1 x') \rangle}_{K'}. \quad (52)$$

More specifically, we study the **condition number** and **eigenvalue decay rate (EDR)** of the kernel as follows.

- **Eigenvalue gap of eNTK**. We start by approximating the sinusoidal eNTK as a Gaussian kernel (Rahimi & Recht, 2007; Mehta et al., 2021), which allows for an analytic form of the spectral decomposition. Using this approximation, we can also estimate the EDR. Here, we observe that an increase in the scaling factor $\alpha$ directly leads to a decrease in the EDR.

- **Condition number**. Building on recent studies of the condition number of the NTK, which significantly affects the convergence rate of neural networks (particularly in the kernel regime) (Xiao et al., 2020; Liu & Hui, 2023), we demonstrate that WS-SNF approximately achieves a low condition number for the eNTK, with the scaling factor $\alpha$ playing a crucial role due to the lower EDR.

We start with the results from Mehta et al. (2021).

**Theorem 7** (Theorem 1 in Mehta et al. (2021).)**.** *Suppose each entry of $W_1$ is initialized with a Gaussian distribution of mean 0 and variance $\sigma^2$. Then for any $x$ and $x'$, we have*

$$|\mathbb{E}_{W_1}[\langle \sin(W_1 x), \sin(W_1 x') \rangle]| \leq C_1 \exp\left(-\frac{\sigma^2 \|x - x'\|_2^2}{2}\right), \quad (53)$$

Theorem 7 states that the expectation over the first-layer weights can be bounded by Gaussian kernels, where the bandwidth is determined by the initialization variance $\sigma$. The scaling factor in Eq. (52) directly affects the variance term, which can be replaced with $\alpha^2 \sigma^2$, without loss of generality. Note that it can be handled similarly in the case of cosine activation function.

Returning to the eNTK, we start by analyzing $K$. From Eq. (53), the kernel element can be approximated as

$$K(x_i, x_j) \leq C_1 \exp\left(-\frac{\alpha^2 \sigma^2 \|x_i - x_j\|^2}{2}\right) = K^*(x_i, x_j). \quad (54)$$

Assuming $K^* - K \succeq 0$, then $\lambda_i^* \geq \lambda_i$ holds for all eigenvalue indices $i$, where $\lambda_i^*$ and $\lambda_i$ denote the $i$-th eigenvalues of the kernels $K^*$ and $K$, respectively. Next step is to compute the eigenvalues of $K^*$ (*i.e.,* eigenvalue upper bound of the $K$). The eigenvalues of $K^*$ can be represented as (Rasmussen & Williams, 2006):

$$\lambda_i \leq \lambda_i^* = C_1 \sqrt{\frac{2}{1 + 2\alpha^2 \sigma^2 + \sqrt{1 + 4\alpha^2 \sigma^2}}} \left(\frac{2\alpha^2 \sigma^2}{1 + 2\alpha^2 \sigma^2 + \sqrt{1 + 4\alpha^2 \sigma^2}}\right)^i, \quad (55)$$

and $K'$ can be handled similarly. In Eq. (55), we assume that $x \sim \mathcal{N}(0, \sigma_x^2)$, and set $\sigma_x = 1$ without loss of generality. Note that the eigenfunctions of the Gaussian kernel $K^*$ can be computed via Hermite polynomial expansion; however, this does not guarantee characteristics of the eigenfunctions of $K$, so we do not take into account. In Eq. (55), we can compute the eigenvalue decay rate (EDR) from arbitrary index $k$ to $k+1$ by examining the base number. Intuitively, we can see that the larger the $\alpha$, the smaller the EDR. Considering both kernels $K$ and $K'$, characterizing the exact eigenvalue of $K_{\text{NTK}} = K + K'$ is open problem in the field of random matrix theory (Tao, 2012). However, we analyze the spectrum indirectly through Weyl's inequality from Courant-Fischer's theorem (Horn & Johnson, 2012). Let $\lambda_k(K)$ and $\lambda_k(K')$ the $k (\leq n)$-th largest eigenvalue of each kernel. Then:

$$\lambda_k(K') + \lambda_1(K)\rho^{n-1} \leq \lambda_k(K_{\text{NTK}}) \leq \lambda_k(K') + \lambda_1(K). \quad (56)$$

Here, $\rho$ denotes the EDR of $K$. In other words, the bound of $\lambda_k(K_{\text{NTK}})$ is determined by the EDR of $K$, where a lower EDR (*i.e.,* a larger scaling factor) contributes to an increase in the lower bound of the eNTK eigenvalue for each index.

This theoretical analysis aligns with our empirical observations. Specifically, as shown in Fig. 5b, the $x$-axis represents the distribution of normalized eigenvalues. The eigenvalues of the weight-scaled network are more densely distributed compared to the default case (*i.e.,* no weight scaling applied), indicating that the EDR of the weight-scaled network is lower. Intuitively, a smaller EDR corresponds to a smaller condition number, further suggesting that the optimization trajectory of weight scaling is relatively well-conditioned.

Recently, Saratchandran et al. (2024a) theoretically derived the minimum eigenvalue of the eNTK in periodically activated MLPs of general depth. Under certain assumptions, the minimum eigenvalue is shown to be asymptotically bounded, involving the variance of the initialization distribution (specifically, larger the variance, larger the minimum eigenvalue), which also supports our results.

### C.4 Non-kernel approach: analysis via Hessian eigenvalues

Understanding the structure of the Hessian is a crucial aspect in the field of optimization. The Hessian matrix represents the second-order derivatives of the loss function with respect to each parameter, and its eigenvalues provide insights into the local curvature of the model's parameter space. The condition number of the Hessian, defined as $\lambda_{\max}/\lambda_{\min}$, directly affects the convergence rate in gradient-based convex optimization problems, which is provable (Nesterov et al., 2018).

On the other hand, in deep learning, the Hessian is also widely studied to gain a better understanding of the complex optimization trajectories and characteristics of deep neural networks. For instance, Ghorbani et al. (2019) examined the outlier eigenvalues of the Hessian in neural networks (i.e., eigenvalues detached from the main bulk) and demonstrated through comprehensive empirical analyses that these outliers can slow down the network's convergence.

Building on this line of research, in this subsection, we plot the condition number (CN) of the Hessian during training in three different settings. Fig. 10 illustrates that the CN[3] of WS-SNF remains significantly lower than that of other comparison models throughout training. This suggests that WS initialization leads to a better-conditioned Hessian not only in the kernel-approximated regime but also in the nonlinear regime, contributing to more efficient training dynamics.

To provide deeper insights and establish connections with related work on spectral analyses of the Hessian in neural networks, we plot the eigenspectrum snapshot of the Hessian during training in Fig. 11. In both networks, we observe the occurrence of 'outlier eigenvalues' during training. However, the default setting exhibits a much more extreme value (even at initialization), which serves as a proxy for poorly conditioned optimization and slow convergence, aligning with the observations in Ghorbani et al. (2019) and Rathore et al. (2024).

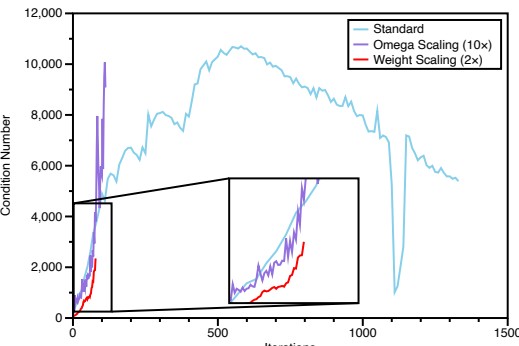

Figure 10. **Condition number of Hessian.** The WS initialization consistently results in a lower condition number compared to other models, indicating that without the approximation of kernelized networks, the optimization path of WS-SNF is well-conditioned.

---

[3]For numerical stability, similar to Section 4.3, we compute the average of the top 3 eigenvalues for $\lambda_{\max}$ and the bottom 3 eigenvalues for $\lambda_{\min}$, where the scale of the eigenvalues are considered in terms of their absolute values. For Hessian computation, we use 'PyHessian' (Yao et al., 2020).

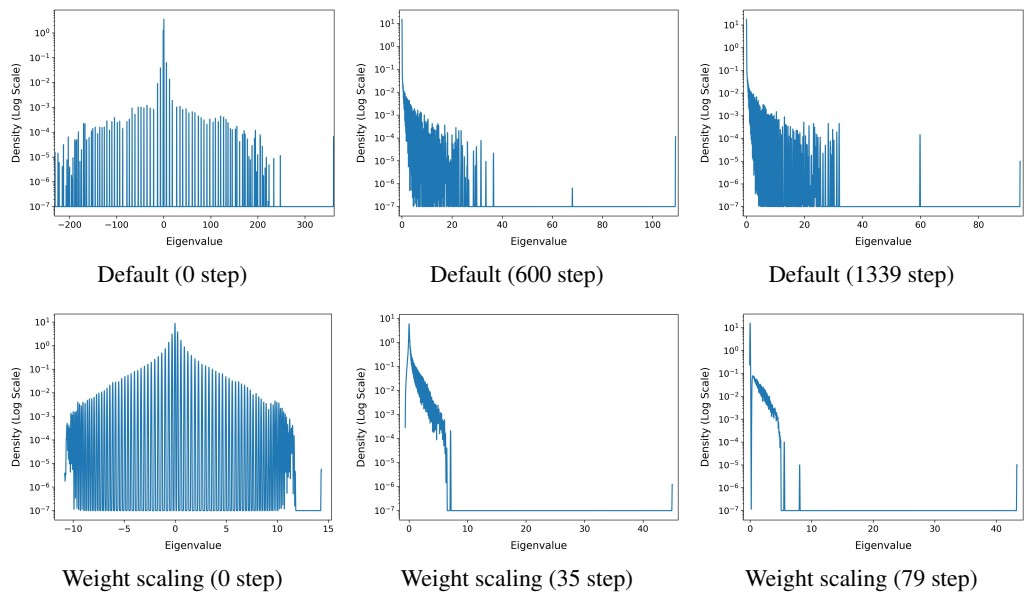

Figure 11. **Eigenspectrum of the Hessian during training**: The first column represents the eigenspectrum at initialization, the second column corresponds to the eigenspectrum at mid-term training, and the third column shows the eigenspectrum at the end of training.

# D    FREQUENCY SPECTRUM OF 2D IMAGES

To further validate our findings, we extend our analysis to Kodak image dataset. Following the methodology of Shi et al. (2022), we partition the frequency map into several discrete bands and compute the mean values within each band.

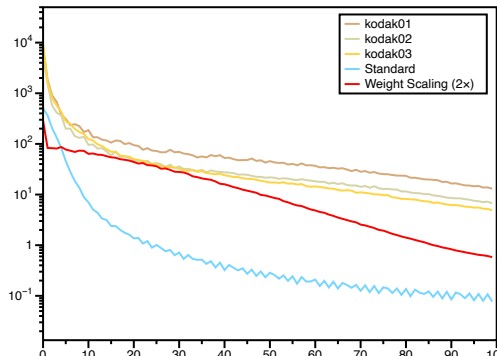

Figure 12. **Frequency distribution of SNFs and 2D signals.** We partition the frequency map into 100 subgroups to facilitate the visualization of 2D signals in a 1D format. Natural images consistently exhibit similar frequency distribution patterns. Notably, WS-SNF demonstrates a better capability in representing high-frequency components, even in its initial state.

Fig. 12 illustrates the frequency distributions across three distinct images from Kodak dataset, as well as the initial output of SNFs. Our analysis reveals that natural images exhibit consistent spectral characteristics across the dataset, providing empirical support for our claim presented in Section 5.2. Moreover, we observe that the initial functional generated by weight-scaled SNF demonstrates a notably higher frequency distribution, consistent with our theoretical analysis.

# E MORE DISCUSSION

## E.1 THE EFFECT OF WEIGHT SCALING IN MINI-BATCH TRAINING

Most of our primary experiments were conducted under a full-batch training to prioritize fast convergence. In this section, we provide supplementary experiments on mini-batch training for image regression to further investigate the effect of weight scaling.

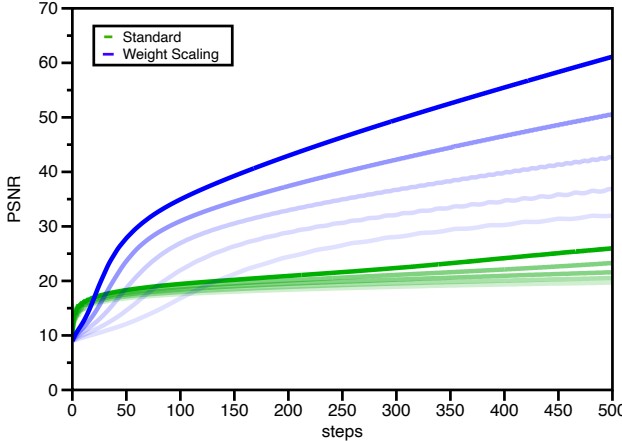

Figure 13. **PSNR curves for mini-batch training.** PSNR curves for default and weight scaled SNFs with various batch sizes. Darker colors represent larger batch sizes.

We trained SNFs on a $512{\times}512$ sized single image using five different batch sizes $\{2^{17}, 2^{16}, 2^{15}, 2^{14}, 2^{13}\}$ over 500 iterations. Fig. 13 demonstrates that weight scaling is *also effective in mini-batch training*, achieving significant acceleration compared to standard SNFs. These results emphasize the robustness and applicability of the proposed method.

E.2 ON THE EFFECT OF WEIGHT SCALING ON OTHER ACTIVATION FUNCTIONS

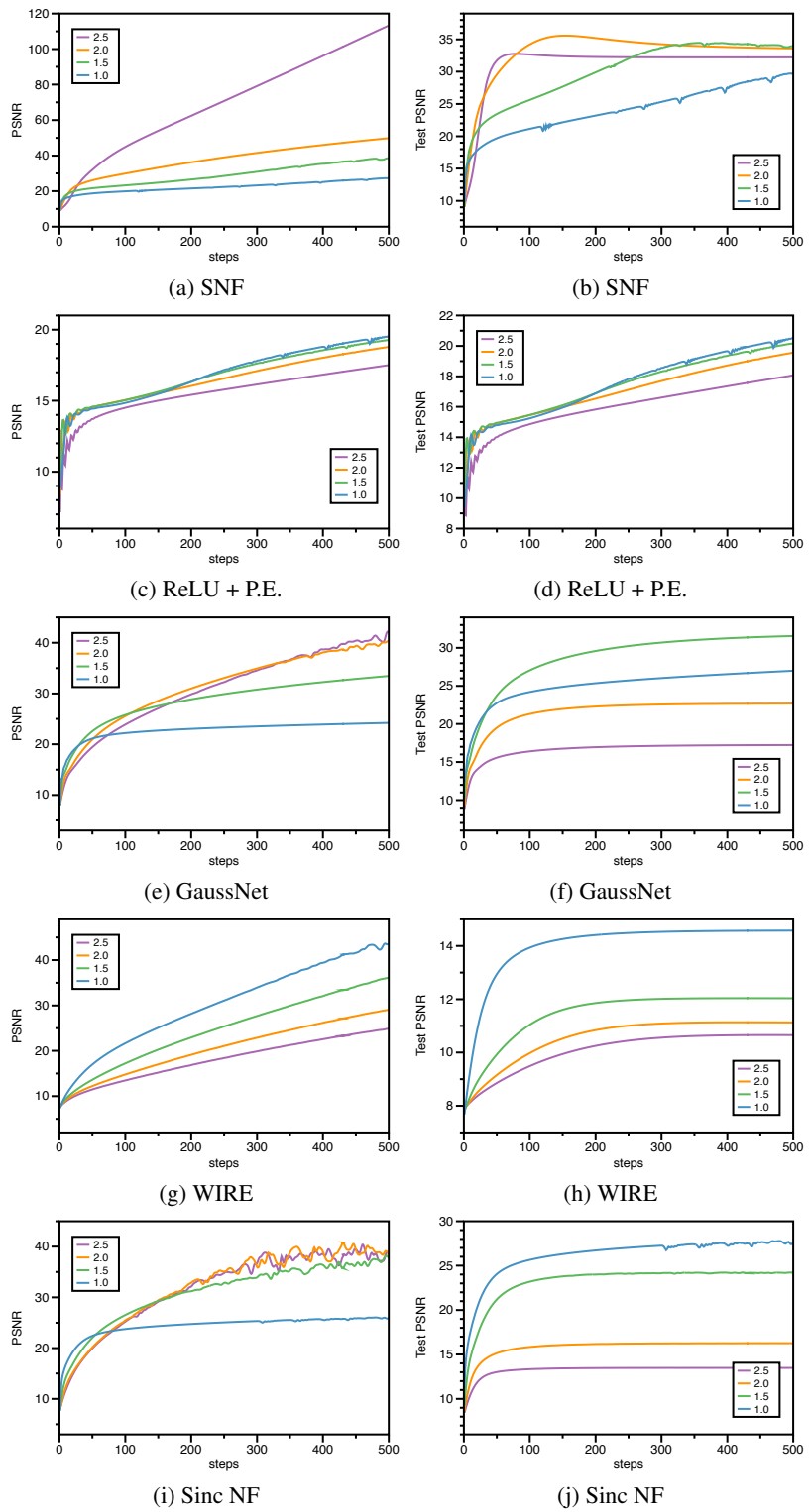

Figure 14. **Weight scaling in various architectures.** WS substantially affects training of the neural field with periodic activations. We present curves for PSNR (first column), as well as Test PSNR (second column).

We extend our investigation to examine the effect of weight scaling on different activation functions. Specifically, we evaluate ReLU, the most commonly used homogeneous activation function, along with other periodic activations beyond sinusoidal. Fig. 14 depicts the train and test PSNR curves over 500 iterations with varying values of $\alpha$. As expected, neural networks with ReLU activations are minimally affected by the scale of weight initialization. In contrast, architectures, such as GaussNet, WIRE, and Sinc NF are significantly influenced by weight scaling. This is because the weight distribution in non-homogeneous networks has a substantial impact on their performance, primarily through its effect on the functional frequency, as discussed in our manuscript.

We observe that weight-scaled GaussNet and Sinc NF also enjoy better performance in terms of the training PSNR, but their test PSNR degrades more severely than in SNFs. Moreover, WIRE shows a degraded performance with weight scaling. While our analysis primarily focused on sinusoidal neural fields, we emphasize that initialization for other periodic activations also impacts acceleration and generalization of neural field training. We leave this as future work, aiming to provide valuable insights into the role of weight initialization in enabling faster convergence.

### E.3 THE OPTIMAL SCALING FACTOR ACCORDING TO THE LEARNING RATE

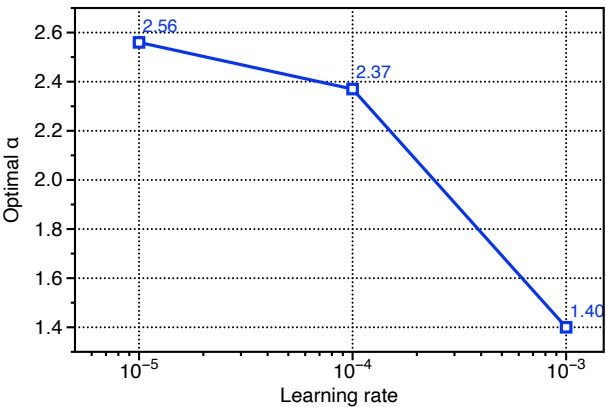

Figure 15. **Optimal scaling factor vs. learning rate.**

We discussed the relationship between the optimal scaling factor and structure properties of neural fields in Section 5.2. However, the learning rate significantly influences training dynamics, potentially leading to instability within the hyperparameter space. Fig. 15 illustrates the optimal weight scaling factor $\alpha$ for different learning rates $\eta \in \{10^{-3}, 10^{-4}, 10^{-5}\}$. Our observations indicate that weight scaling consistently accelerates training across all tested learning rates. Furthermore, a clear trend emerges: smaller values of $\alpha$ are optimal for larger $\eta$. This trend highlights the predictability of the optimal $\alpha$, consistent with other structural properties discussed in our main section.

## F  EXPERIMENTAL DETAILS AND QUALITATIVE RESULTS

### F.1  EXPERIMENTAL DETAILS ABOUT FIG. 2B

Fig. 2b presents the tradeoff between acceleration and generalization when tuning $\omega$ or $\alpha$. Therefore, we choose the ranges of $\omega$ and $\alpha$ that can help us capture the full tradeoff curve. Specifically, for frequency tuning, we explore a broad range, $\omega \in [30, 6510]$, with intervals of 270, while also including smaller values ($\omega = 10$ and 20) in the search space. On the other hand, weight scaling requires a narrower search range, with $\alpha \in [0.4, 40]$ with the grid 0.2.

### F.2  EXPERIMENTAL DETAILS ABOUT FIG. 3

For the $k$-th layer of an $l$-layer SNF, weight scaling increases the gradient magnitude by $\alpha^{l-k}$ compared to standard setting. By scaling down the learning rate of each layer by $\eta \cdot \alpha^{l-k}$, the gradient update magnitudes become similar to those of the default while preserving the initial functional of weight scaling (Fig. 3b). In contrast, for Fig. 3c, the learning rate of $k$th layer is amplified by $\eta \cdot \alpha^{l-k}$ to only utilize the effect of larger gradients. This is inspired by our analytic derivations at the end of Section 4.1 (and Appendix B.3).

### F.3  EXPERIMENTAL DETAILS ABOUT EQ. (10)

We clarify that the optimization Eq. (10) is solved by a simple grid search. That is, we have trained SNFs with many different values of $\alpha$ and selected the value which maximizes the training speed and meets the constraint. We have tried all $\alpha$ within the range $[1.0, 4.0]$ with the grid size 0.2.

### F.4  TRAINING SETTINGS AND BASELINES

In this section, we provide detailed information about our experiments. For data fitting tasks, we used the Adam optimizer (Kingma, 2015) with a learning rate of 1e-04 (except spherical data, in this case we use a learning rate of 1e-05), without any learning rate scheduler, except for the occupancy field experiments (in which case, we use PyTorch learning rate scheduler). Each experiment was conducted with 5 different seeds, and we report both the average and standard deviation of the evaluation metric. We note that all baselines in this work are MLP architectures, with slight modifications to the input features or activation functions. We used NVIDIA RTX 3090/4090/A5000/A6000 GPUs for all experiments. Additionally, we set the hyperparameters of each network using a grid search, as shown in Table 2.

**ReLU + P.E.** (Mildenhall et al., 2020): For this architecture, we use fixed positional encoding applied to the input $x$, with $K$ frequencies that lift the dimension of the inputs, i.e., $\gamma(x) = \left(\sin(2^0\pi \cdot x), \cos(2^0\pi \cdot x), \ldots, \sin(2^{K-1}\pi \cdot x), \cos(2^{K-1}\pi \cdot x)\right)$.

**FFN** (Tancik et al., 2020): For this architecture, we use Gaussian positional encoding applied to the input $x$, i.e., $\gamma(x) = [\sin(2\pi G \cdot x), \cos(2\pi G \cdot x)]$, where $G$ defines the variance of the Gaussian distribution.

**MFN** (Fathony et al., 2021): For this architecture, we use MLP with multiple Gabor filters, which achieves superior results than Fourier filter networks (Fathony et al., 2021).

**GaussNet** (Ramasinghe & Lucey, 2022): For this architecture, we use the Gaussian activation function, i.e., $\sigma(Z) = \exp\left(-(s \cdot Z)^2\right)$, where $Z$ denotes the pre-activation matrix and $s$ is a hyperparameter that defines the bandwidth of the Gaussian function.

**WIRE** (Saragadam et al., 2023): For this architecture, we use a Gabor wavelet-based activation function, i.e., $\sigma(Z) = \exp(j\omega \cdot Z)\exp(-(s \cdot Z)^2)$, where $\omega$ and $s$ are hyperparameters.

**SIREN** (Sitzmann et al., 2020b): We use the sine activation function $\sigma(z) = \sin(\omega \cdot Z)$, where $\omega$ in the first layer is a tunable hyperparameter.

**Neural field scaling law (NFSL)** (Saratchandran et al., 2024b): We modify the initialization distribution of the last layer of SIREN, i.e., $W^{(L)} \sim \mathcal{U}\left(-\sqrt{6}/\left(n_{L-1}^{3/4}\omega\right), \sqrt{6}/\left(n_{L-1}^{3/4}\omega\right)\right)$, where $n_{L-1}$ denotes the width of the $(L-1)$-th layer.

**Sinc NF** (Saratchandran et al., 2024c): In this case, we use activation function as a sinc function, *i.e.,* $\sigma(Z) = \sin(\omega x)/\omega x$, where $\omega$ is the fixed parameter. We use $\omega = 8$ for all experiments, except for audio fitting (in this case, we use $\omega = 16$).

**Xavier uniform initialization** (Glorot & Bengio, 2010): In this case, we use the standard Xavier uniform initialization for the SNF architecture, i.e., $W^{(l)} \sim \mathcal{U}\left(-\sqrt{6}/\sqrt{n_{l-1} + n_l}, \sqrt{6}/\sqrt{n_{l-1} + n_l}\right)$, $l \in [1, L]$. We note that the Xavier uniform initialization in SNF does not yield reasonable results. However, the purpose of presenting it is to highlight the *initialization principle of SNFs* (Appendix A), which guarantees stable training.

Table 2. **Detailed information about hyperparameters.** We provide the exact hyperparameter settings for each domain in the table below. 'default' in ReLU+P.E. denotes the using nyquist sampling methods (Saragadam et al., 2023).

| | ReLU+P.E. | FFN | | SIREN | GaussNet | WIRE | | Weight scaling |
|---|---|---|---|---|---|---|---|---|
| | k | $\sigma$ | m | $\omega$ | $s$ | $\omega$ | $s$ | $\alpha$ |
| Image | 10 | 10 | 256 | 30 | 10 | 20 | 10 | 2.37 |
| Occ. Field | default | 2 | 10 | 10 | 10 | 10 | 40 | 3.7 |
| Spherical Data | default | 2 | 256 | 30 | 10 | 10 | 20 | 2.5 |
| Audio | default | 20 | 20 | 3000 | 100 | 60 | 10 | 2.0 |

## F.5 NEURAL DATASET EXPERIMENTS

In recent research on constructing large-scale NF datasets (Ma et al., 2024; Papa et al., 2024), there has been growing interest in this area. However, no studies have investigated how initialization schemes affect the overall process, from constructing SNF-based neural datasets to their application in downstream tasks. To address this gap, we first measure the time required to fit the entire dataset to NFs and then report the classification accuracy on the neural dataset while varying the scale factor $\alpha$. The results are shown in Table 3.

**Analysis.** Interestingly, despite the efficiency of WS-SNFs in terms of training time (*i.e.,* the time required to fit each datum into an individual NF until reaching the target PSNR), the test accuracy (*i.e.,* classification accuracy on the test set SNFs), specifically for the case of $\alpha = 1.5$, does not significantly decrease. Moreover, we observed that, in the case of $\alpha = 2.0$, test accuracy dropped by an average of 5 percentage points across all PSNR values. This mirrors a similar phenomenon discussed in Papa et al. (2024), where constraining the weight space of NFs was identified as a key factor in downstream task performance. More precisely, scaling initialization distributions causes the neural functions to lie in a larger weight space, making it harder to learn features from the weights. Furthermore, we propose an interesting research direction: *can we create a neural dataset quickly without compromising the performance of downstream tasks and without incurring additional cost?*

**Experimental details.** For training, we used 'fit-a-nef' (Papa et al., 2024), a JAX-based library for fast construction of large-scale neural field datasets. We used the entire dataset for NF dataset generation (*i.e.,* 60,000 images for MNIST and CIFAR-10, respectively). Additionally, we used an MLP-based classifier targeted at classifying each NF's class, which consists of 4 layers and a width of 256, and was trained with cross-entropy loss for 10 epochs.

Table 3. **Neural dataset experiments.** Each neural dataset is trained until it reaches the batch-averaged **target PSNR**. The **training time** refers to the total time required to construct the entire INR dataset. A classifier is then trained on the neural dataset, and the resulting **test accuracy** is reported for each configuration. We use SIREN as the NF architecture, which is a widely used baseline in this area.

| | | Neural MNIST | | Neural CIFAR-10 | |
|---|---|---|---|---|---|
| | Target PSNR | Training time | Test accuracy | Training time | Test accuracy |
| | 35 | 14m 23s | 97.95±0.00 | 19m 14s | 47.98±0.01 |
| SIREN | 40 | 18m 20s | 97.76±0.00 | 29m 45s | 48.46±0.01 |
| | 45 | 23m 23s | 97.53±0.01 | 47m 54s | 48.09±0.01 |
| | 35 | 7m 22s | 97.56±0.00 | 10m 52s | 47.15±0.02 |
| WS (× 1.5) | 40 | 8m 22s | 97.81±0.00 | 17m 21s | 47.72±0.01 |
| | 45 | 10m 41s | 97.02±0.01 | 25m 52s | 46.08±0.03 |
| | 35 | 5m 25s | 97.48±0.01 | 8m 26s | 43.30±0.02 |
| WS (× 2.0) | 40 | 5m 32s | 97.06±0.01 | 9m 20s | 42.86±0.02 |
| | 45 | 5m 41s | 97.49±0.00 | 10m 51s | 42.70±0.01 |

### F.6 NEURAL RADIANCE FIELDS EXPERIMENTS

In this subsection, we provide details and quantitative results of novel view synthesis via neural radiance fields (NeRFs) (Mildenhall et al., 2020). Neural radiance fields (NeRFs) represent scenes using neural fields. More specifically, NeRF employs neural fields (*i.e.,* MLPs) with finite amount of training data to capture the scene, aiming to recover the scene in a continuous manner. At the end, we input unseen coordinates into the model and evaluate the test PSNR.

**Datasets.** We use the 'Lego' and 'Drums' data from the 'NeRF-synthetic' dataset (Mildenhall et al., 2020), which is publicly available online. Each dataset contains 100 training images, 100 validation images, and 200 test images, along with their corresponding camera directions.

**Implementation details.** We mainly follow the settings of WIRE (Saragadam et al., 2023), using the 'torch-ngp' codebase. Specifically, we use a 4-layer volume density network and a 4-layer color network, each with a width of 182, except for WIRE. This exception is due to the complex-valued weights of WIRE, where we use a width of 128 for fair comparison. For the WS network, we apply weight scaling only to the volume density network.

**Task 1: training with 800×800 images.** In this case, we use $\omega_0 = 15$ and $\omega_h = 5$ for the SNF family, frequency lifting factor $k = 10$ for ReLU + P.E., $s = 20$ for GaussNet, and $\omega = 5, s = 10$ for WIRE. Additionally, we use positional encoding scheme only for ReLU networks. We use learning rate $1e - 02$ for ReLU+P.E., $1e - 03$ for FFN, $3e - 04$ for SNF family, $1e - 03$ for GaussNet, and $6e - 04$ for WIRE. We train all models until 300 epochs. We train the model with 100 images and evaluate the test PSNR with 100 untrained images.

**Task 2: training with 200×200 images, with longer epochs.** Additionally, as in Saragadam et al. (2023), we train model with $200 \times 200$ images, and trained until further epochs (600 epochs). In this case, we use $\omega_0 = 15$ and $\omega_h = 3$ for the SNF family, frequency lifting factor $k = 10$ for ReLU + P.E., $s = 20$ for GaussNet, and $\omega = 20, s = 40$ for WIRE. Also, we use positional encoding scheme only for ReLU networks. We use learning rate $6e - 03$ for ReLU + P.E., $3e - 03$ for FFN, $9e - 04$ for SNF family, $1e - 03$ for GaussNet, and $9e - 04$ for WIRE. We train the model with 100 images and evaluate the test PSNR with 100 untrained images.

**Results.** We report the test PSNR and learning curve in Table 4 and Fig. 16 respectively. For both datasets and both training resolutions, WS achieves a higher test PSNR compared to other baselines. We emphasize that WS primarily focuses on fast training of neural fields, but here we also demonstrate its good generalization quality. Qualitative results can be found in Fig. 21.

| Method | 800×800 (300 Epochs) | | 200×200 (600 Epochs) | |
|---|---|---|---|---|
| | Lego | Drums | Lego | Drums |
| ReLU + P.E. (Mildenhall et al., 2020) | 26.02 | 20.97 | 31.57 | 26.21 |
| SIREN (Sitzmann et al., 2020b) | 27.68 | 24.05 | 32.02 | 27.73 |
| FFN (Tancik et al., 2020) | 26.96 | 22.10 | 30.40 | 24.68 |
| GaussNet (Ramasinghe & Lucey, 2022) | 25.95 | 22.42 | 30.61 | 26.24 |
| WIRE (Saragadam et al., 2023) | 26.71 | 23.62 | 32.09 | 27.73 |
| NFSL (Saratchandran et al., 2024b) | 27.79 | 24.04 | 32.10 | 27.64 |
| Weight Scaling (ours) | **28.17** | **24.10** | **32.48** | **27.86** |

Table 4. **Quantitative results**: test PSNR for NeRF experiments.

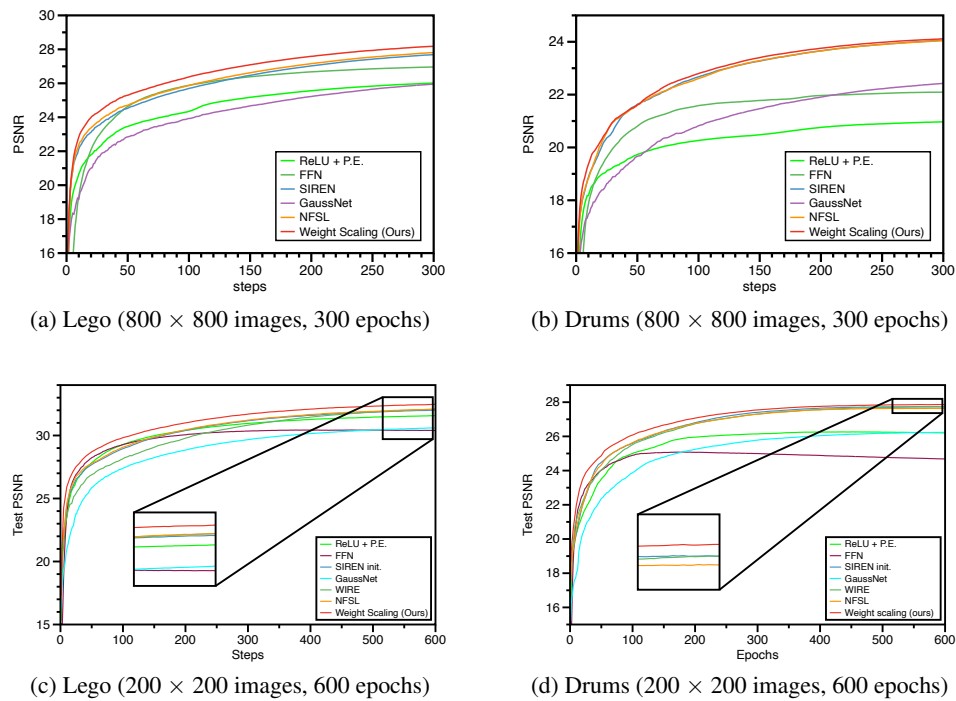

Figure 16. Test PSNR curves for NeRF experiments under different settings. The first row represents experiments conducted with $800 \times 800$ images over 300 epochs, while the second row shows experiments conducted with $200 \times 200$ images over 600 epochs. WS initialization consistently achieves better test PSNR compared to the baselines.

### F.7 SOLVING DIFFERENTIAL EQUATIONS WITH PARTIAL OBSERVATIONS

In this subsection, we provide details about solving wave equation: a type of second order time-dependent partial differential equation (PDE) describing wave propagation in time $t$. The initial conditions are only the given information, neural fields aim to extrapolate the equation in timescale domain. We use the same formulation as in Sitzmann et al. (2020b).

**Datasets and training details.** We set $f(x)$ (*i.e.,* neural network) to reconstruct and extrapolate Gaussian pulse wave equation, same setup as in Sitzmann et al. (2020b). Input coordinates are 3-dimensional (*i.e.,* 2-dimensional spatial and 1-dimensional temporal timescale from $t = 0$ to 4), and outputs 1-dimensional Gaussian pulse value for each spatiotemporal coordinate. Other hyperparameters are set the same as in Sitzmann et al. (2020b), except for the number of iterations (we trained for up to 40k iterations).

**Results.** We present the qualitative results in Fig. 22. However, in Fig. 22, the WS network consistently predicts the boundary of the Gaussian, compared to other architectures[4]. Qualitative comparisons are provided using coordinate-wise error (*i.e.,* MSE) at $t = 0$. It can be seen that WS-SNF not only faithfully reconstructs the given observation but is also able to generalize accurately to further timesteps. For WIRE and MFN, they failed to reconstruct the pulse; therefore, we do not provide the results.

---

[4]Since the ground truth Gaussian pulses for $t > 0$ are unavailable, they can only be reconstructed using a PDE solver or utilizing finite element method.

## F.8 Qualitative Results

In this subsection, we provide qualitative results of experiments in Section 5.1. We do not provide reconstructed results for occupancy field reconstruction with Xavier initialization case due to training failure.

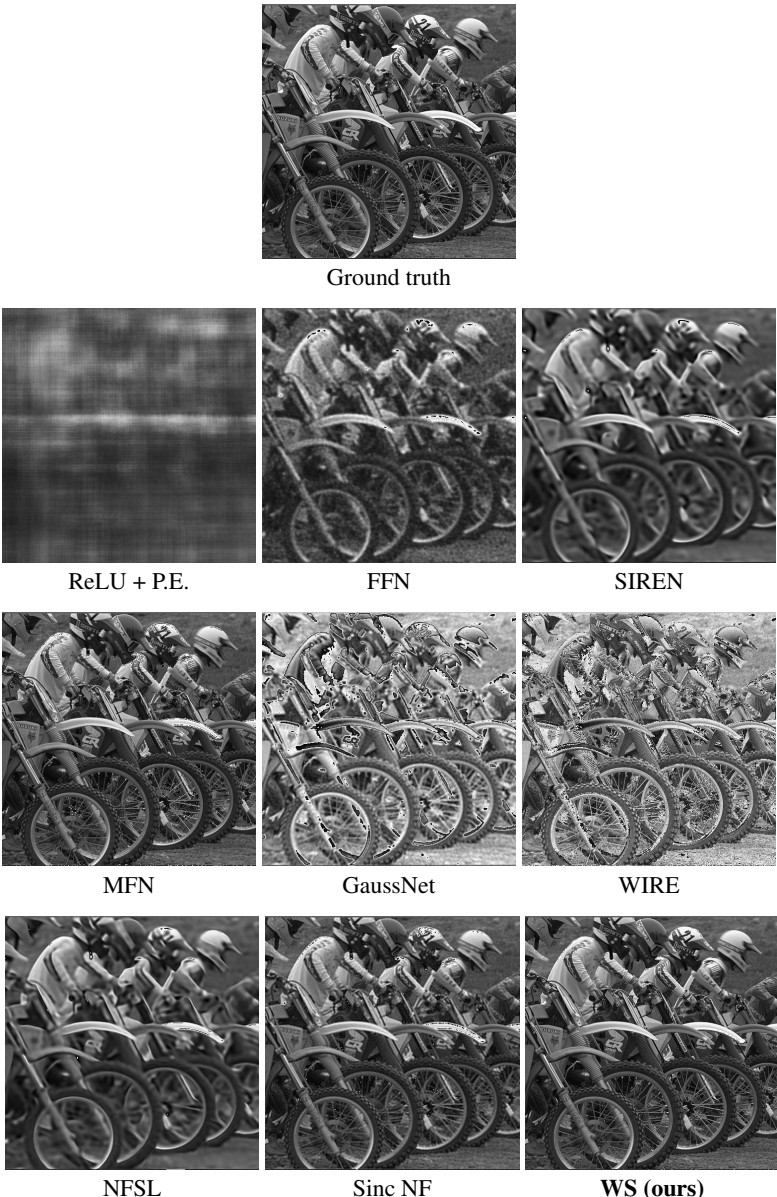

Figure 17. **Image reconstruction**: qualitative results.

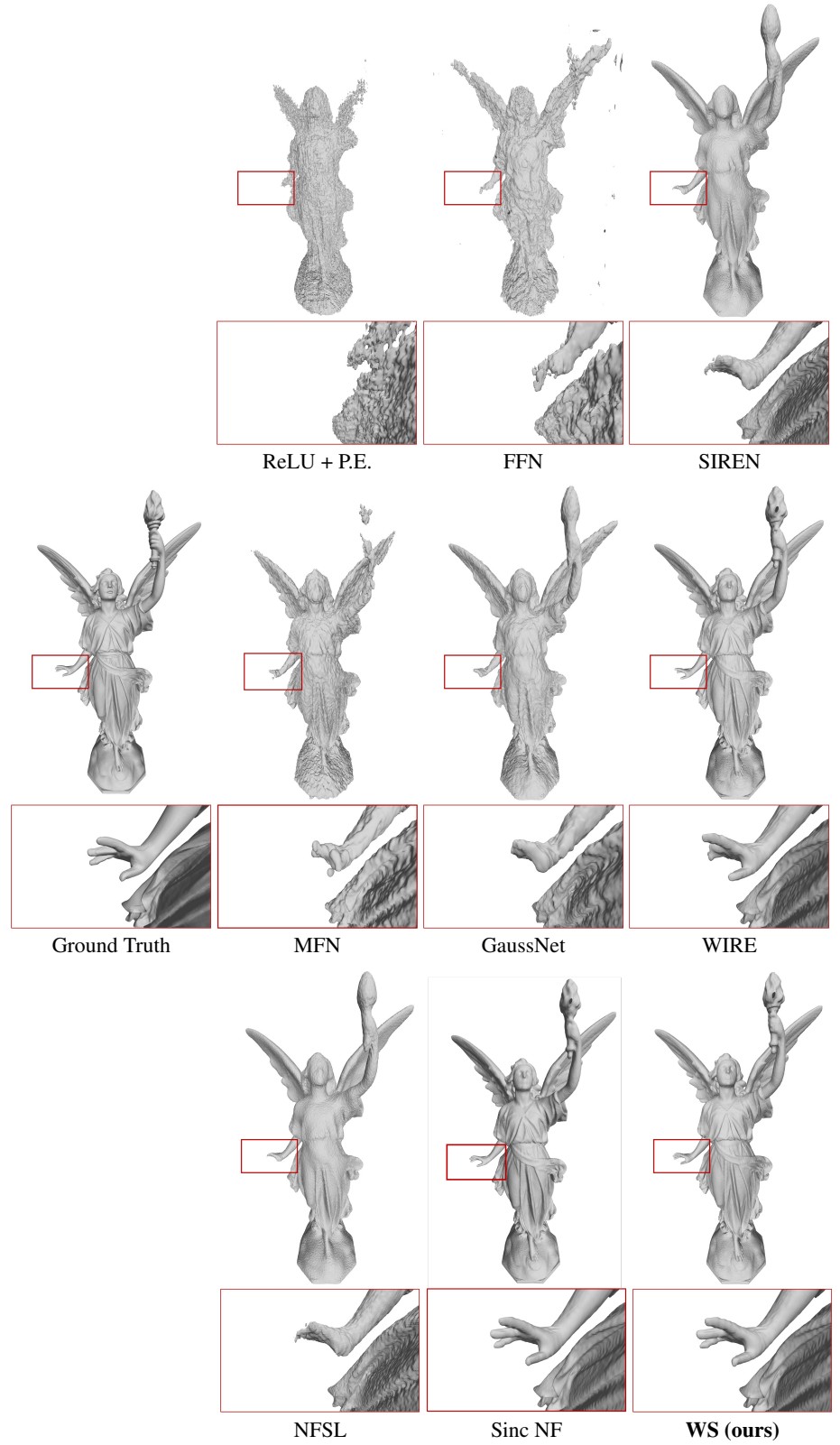

Figure 18. **Occupancy field experiments**: qualitative results.

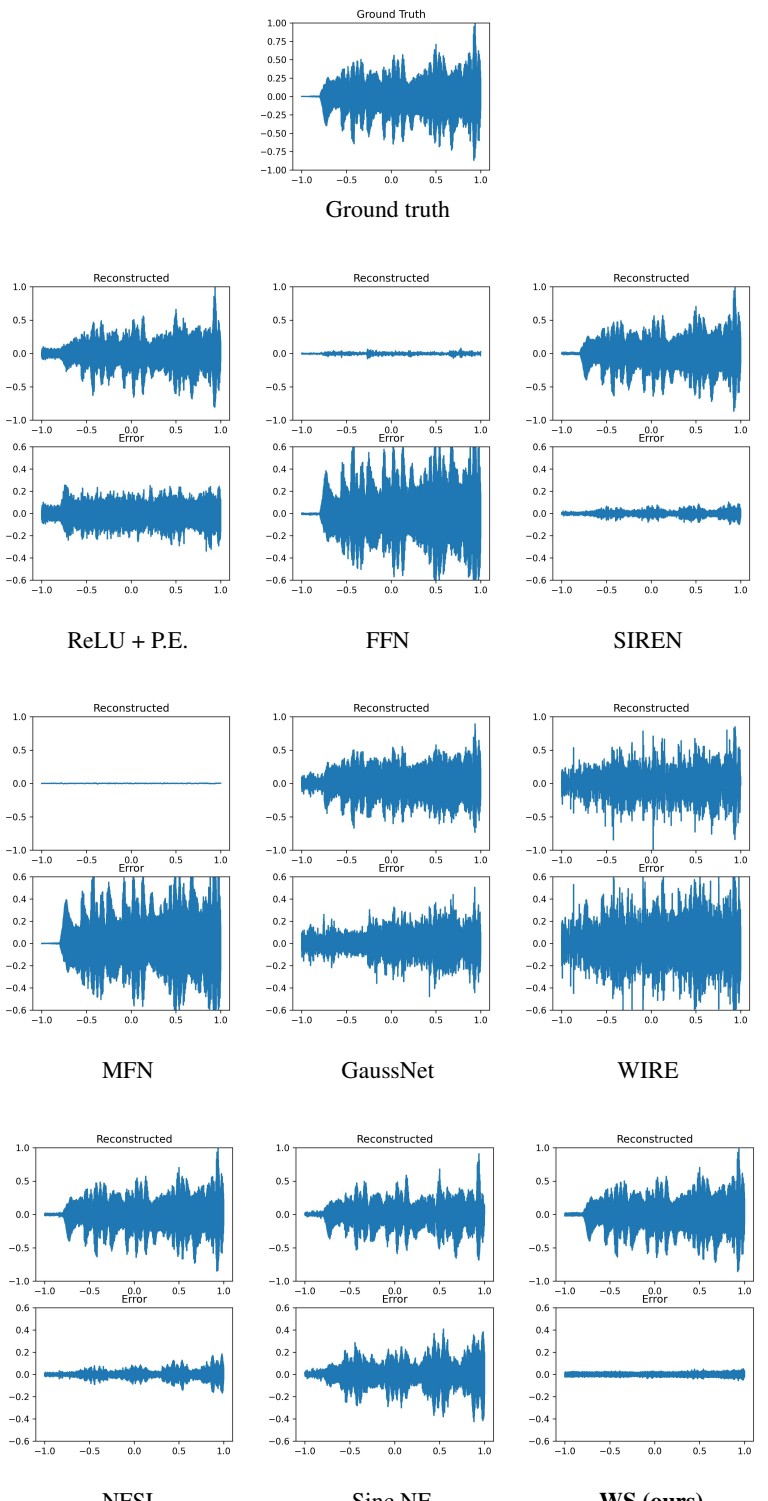

Figure 19. **Audio reconstruction**: qualitative results.

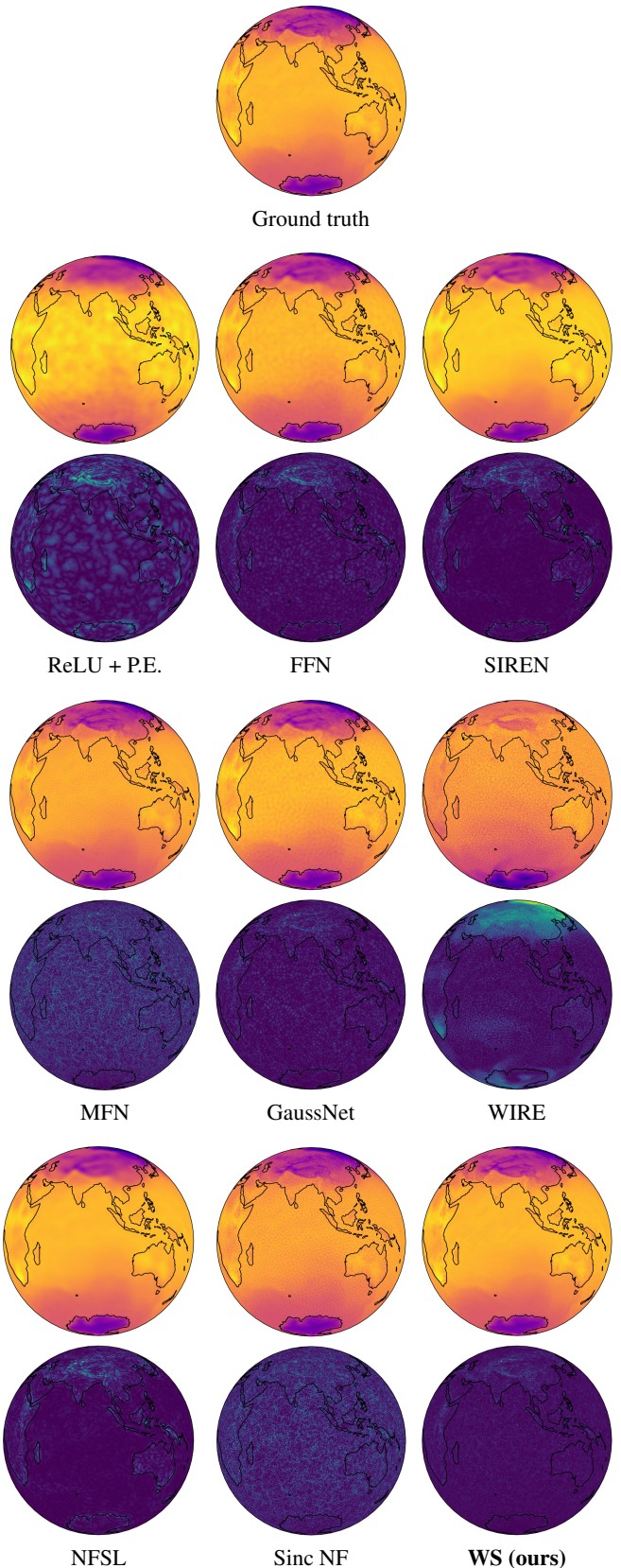

Figure 20. **Spherical data reconstruction**: qualitative results. The top figure of each index shows the reconstructed results, while the bottom figure of each index shows the error map.

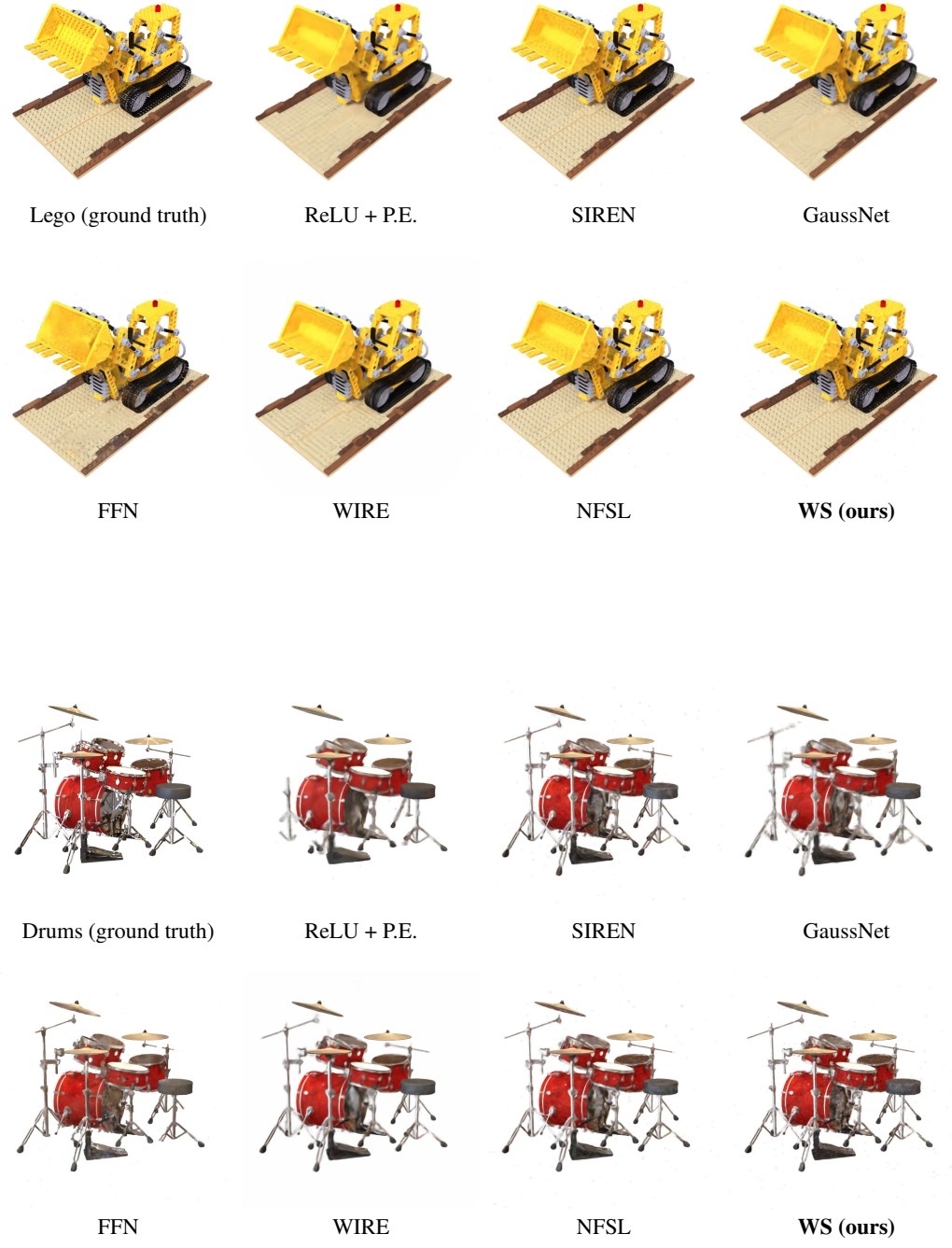

Figure 21. **Novel view synthesis**: qualitative results ($800 \times 800$ images).

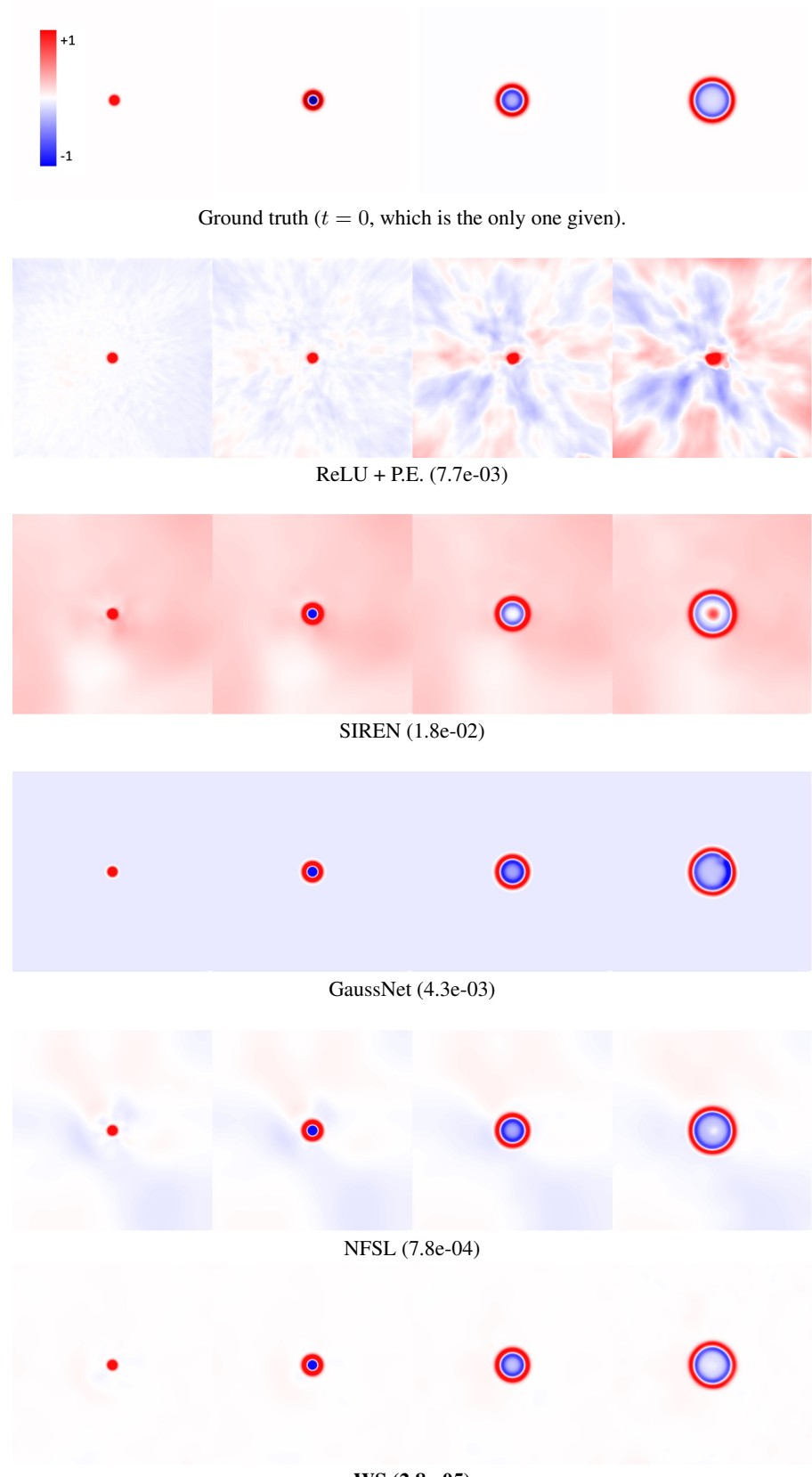

Figure 22. **Solving PDE from partial observation**: qualitative results. Each column represents $t = 0$, 0.07, 0.14, and 0.21 from left to right. The parentheses in the caption* represent the MSE at $t = 0$.

## F.9 TRAINING FOR SIGNIFICANTLY LARGER ITERATIONS

In this section, we present the learning curves for Kodak image dataset over significantly longer iterations. We report the average train and test PSNR values for 5000 training steps, in which most methods converge to saturation. The learning rate scheduler is employed, resulting in improved performance across most methods under extended training setting, and the learning rates are tuned for each architecture to work well with a scheduler.

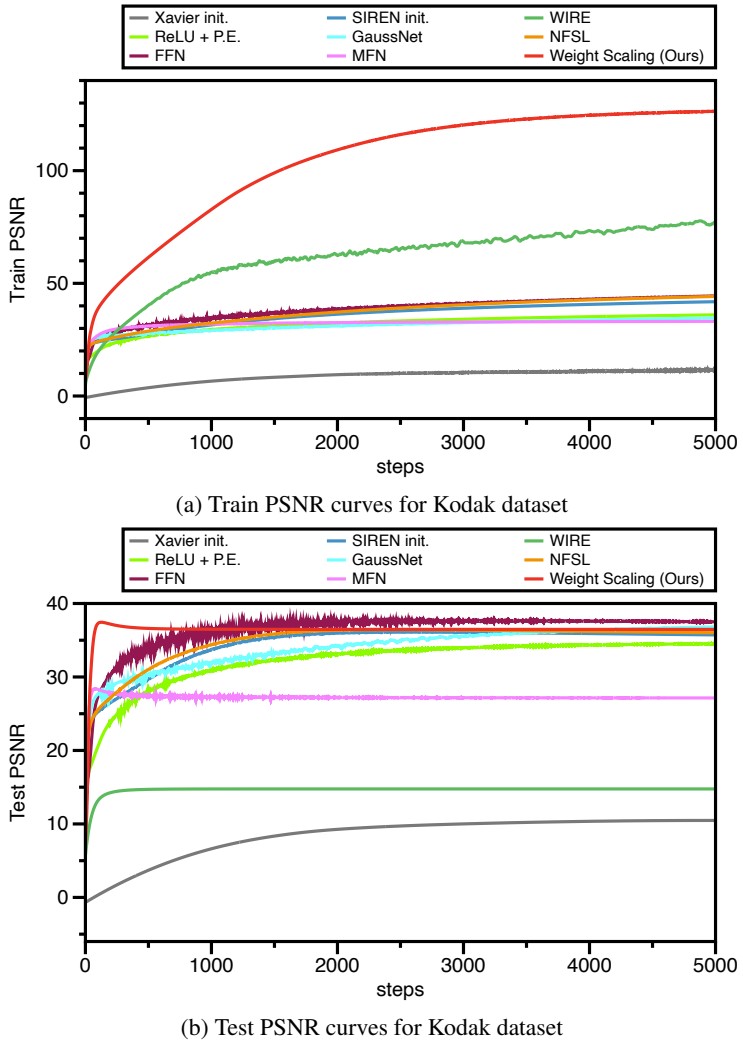

(a) Train PSNR curves for Kodak dataset

(b) Test PSNR curves for Kodak dataset

Figure 23. **PSNR curves over 5000 iterations.** We compared the PSNR of weight scaling to other baselines.

Weight scaling demonstrates superior train accuracy, achieving higher PSNR with substantial acceleration. Moreover, even at saturation, weight scaling outperforms other methods, with the highest PSNR. Notably, our methods maintain its generalization ability even for extended training. Weight scaling reaches its peak test PSNR (37.45 dB), which is comparable to or exceeds that of other methods, much faster, demonstrating the fastest convergence in both train and test performance.

We utilize learning rate $1e-02$ for Xavier initialized SNF, $1e-03$ for ReLU + P.E., $1e-03$ for FFN, $1e-04$ for SNF family, $1e-03$ for GaussNet, $1e-02$ for MFN, and $1e-04$ for WIRE, with Adam optimizer.

