# OpenReview forum: "Fast Training of Sinusoidal Neural Fields via Scaling Initialization"
_ICLR.cc/2025/Conference — ICLR 2025 Poster_

### Official Review · Reviewer_jBfM · 2024-10-26

**Soundness:** 3
**Presentation:** 3
**Contribution:** 2
**Rating:** 6
**Confidence:** 5

**Summary:**

This paper presents a study on sinusoidal neural fields (SNFs) and introduces "weight scaling," a method that accelerates training by adjusting the initialization of each layer, except the final one. The key idea behind weight scaling is straightforward: each layer’s weights are initialized by multiplying with a specific scale factor, which results in faster training across various neural field tasks. The authors benchmark their approach against existing methods, including the original SNF initialization by Sitzmann et al., and demonstrate that weight scaling consistently achieves faster convergence while maintaining robust training and test performance. Furthermore, the authors provide theoretical analysis explaining why weight scaling works.

**Strengths:**

**Originality:** The paper introduces an original idea that I have never seen used before in the neural field literature. Furthermore, the authors take the time to give a good theoretical analysis of why weight scaling works and how it relates to other initialization methods within the literature.

**Quality:** The paper has both theoretical and experimental contributions making it a good quality paper written by the authors. The theoretical contributions are sound and the experiments are compared with recent literature.

**Clarity:** The paper is in general well written. The authors take the time to exposit the main ideas behind weight scaling well and explain their key contributions in a readable manner. I have checked their mathematical proofs in the appendix and they were all well written and explained well.

**Significance:** The paper gives a new way to initialize sinusoidal neural fields which I believe is interesting to the community of researchers working in neural fields. The initialization of neural fields in general is not as commonly studied as other deep networks such as CNNs or Transformers and apart from a few recent works on the topic, I believe this paper offers a nice viewpoint in how sinusoidal neural fields should be initialized for efficient training.

**Weaknesses:**

**Novelty:** My main issue with the paper and what I feel is its main weakness is the fact that it is entirely focused on one type of neural field, namely sinusoidal neural fields. The paper does not say anything about other neural fields in general. The authors do say that their methods would not work on ReLU based neural fields due to ReLU having the positive homogeneity condition and
while the authors do say that sinusoidal neural fields are used within the community I would argue that there are several papers showing that sinusoidal neural fields are not as effective as other neural fields, such as [1], [2], [3], [4].  While the paper is written well I do feel that it being entirely focused on one type of neural field means it won't be as useful to many researchers in the neural fields community.

**Large Scale Experiments:** The paper compares its methodology of weight scaling on a tasks such as image regression, binary occupancy, spherical data, audio data and fit a nef (in the appendix). Yet I noticed there were no experiments on Neural Radiance Fields (NeRFs) which I found very surprising. The authors say that their method is very effective at reducing the training time of neural fields which is generally large yet many of the tasks such as image regression, audio, occupancy are not considered big tasks within the community. On the other hand NeRF is a big experiment that takes anywhere between 4 hours to 12 hours to train, depending on the fidelity the user wants. I would have liked to see how their method did on such a task that requires hours of training. I also looked at the papers the authors compare to namely [1], [2], [3] and noticed that these papers compared their methods on NeRF. In the appendix the authors do compare their method on "fit a nef" on a large scale dataset but these takes less than 1 hour to train even with a SIREN. Thus I feel another big weakness of the paper is that the authors did not compare their method on a task that takes hours to train such as NeRF, so that we can see how their method does on a more complicated large scale neural field.

[1] Ramasinghe et al: Beyond Periodicity: Towards a unifying framework for activations in coordinate-MLPs; ECCV 2022.

[2] Saragadam et al: WIRE: Wavelet implicit neural representations; CVPR 2023.

[3] Saratchandran et al: Scaling insights for optimizing neural fields: CVPR 2024.

[4] Saratchandran et al: A Sampling Theory Perspective on Activations for Implicit Neural Representations: ICML 2024.

**Questions:**

1). Your paper is entirely focused on neural fields that use a sinusoidal activation. The weight scaling method you propose scales the initialization of a sinusoidal neural field by a certain factor $\alpha$, as per eqn (6) in your paper. This method takes the SIREN initialization as its basis. Could you explain what happens if we take a standard initialization scheme like Kaiming uniform or Xavier uniform and scale it in exactly the same way as your eqn. (6)? Does this lead to anything new? Why did you apply your weight scaling to only the SIREN initialization?

2). Could you explain whether your results would go through for other activations that are used within the neural fields literature such as Gaussian [1], Wavelet [2], Sinc [4]?

3). Did you try weight scaling on a NeRF task? I am very surprised there are no NeRF results as this a big experiment that requires hours to train and I noticed some of the papers you are comparing against [1], [2], [3] compare to NeRF. Could you explain whether weight scaling would or wouldn't work on NeRF and if it cannot be made to work on NeRF then why? I think it would really help your paper if you had NeRF results and you were able to show large speed ups on this large scale experiment.

4). Your weight scaling essentially just scales the initialization of the inner layers of the initialization by Sitzmann et al for SIRENs. However, if you look at the denominator of eqn. (3), line 194, there is the term $\omega_h$ which is supposed to be the frequency of the sinusoid being used on that layer. Could we not get the same effect as your initialization by using a sinusoid with frequency
$\frac{\omega_h}{\alpha}$?

5). In appendix C.3 you analyse the condition number of the Hessian of the loss landscape. You mention that the condition number of the Hessian directly affects the convergence rate. However, my understanding is that this is only true for convex loss landscapes. Loss functions associated to neural networks are almost always non-convex, as they employ non-linearities, therefore in such cases the condition number of the Hessian may not be well defined, as the Hessian may have negative eigenvalues or zero as an eigenvalue. Could you comment on this? I don't understand this analysis you have provided in your paper.

--------------------------------------------------------------------------------------------------------------------------------------------------------------------------------
[1] Ramasinghe et al: Beyond Periodicity: Towards a unifying framework for activations in coordinate-MLPs; ECCV 2022.

[2] Saragadam et al: WIRE: Wavelet implicit neural representations; CVPR 2023.

[3] Saratchandran et al: Scaling insights for optimizing neural fields: CVPR 2024.

[4] Saratchandran et al: A Sampling Theory Perspective on Activations for Implicit Neural Representations: ICML 2024.

---

> ### Author Response · Authors · 2024-11-21
>
> Dear Reviewer jBfM,
>
> We thank you for your concise and constructive suggestions, as well as the effort you put into reviewing our work. Below, we provide responses to the weaknesses and questions you mentioned.
>
> ---
> ### **W1.** About the novelty issue.
> > **Novelty**: My main issue with the paper and what I feel is its main weakness is the fact that it is entirely focused on one type of neural field, namely sinusoidal neural fields. The paper does not say anything about other neural fields in general. The authors do say that their methods would not work on ReLU based neural fields due to ReLU having the positive homogeneity condition and while the authors do say that sinusoidal neural fields are used within the community I would argue that there are several papers showing that sinusoidal neural fields are not as effective as other neural fields, such as [1], [2], [3], [4]. While the paper is written well I do feel that it being entirely focused on one type of neural field means it won't be as useful to many researchers in the neural fields community.
>
>
> We respectfully disagree with the reviewer’s point that focusing on sinusoidal neural fields significantly undermines the usefulness and the impact of our work. The reason is threefold.
> - **When initialized with WS, SNF outperforms [1,2,3,4].** It is true that [1,2,3,4] demonstrates that neural fields with other activations outperform SIREN-initialized SNFs. However, our experimental results suggest that the proposed weight-scaled initialization makes the SNFs great again, making it outperform the GaussNet [1], WIRE [2], NFSL [3], and Sinc [4] baselines quite consistently over various setups (Table 1); we have compared with [4] in our response below. Thus, we argue that the claim ‘sinusoidal neural fields are not as effective as other neural fields’ is no longer true.
> - **SNFs are widely used in practice.** We find that SNFs are still widely used in various domains, e.g., [5,6,7,8,9]. In fact, the authors of [1,2,3,4] still utilize sinusoidal activations in their recent work [10]. Thus, we believe that developing an efficient training scheme for SNFs can make a significant practical impact.
> - **Our work reveals the intimate relationship between the initialization and training cost.** The initialization scale of neural nets have been actively studied in both practical and theoretical domains. However, the connection between the training efficiency and initialization is still poorly understood. We strongly believe that our case study on sine-activated networks, which demonstrates that better initialization can greatly improve the training efficiency for free, can inspire much future effort in this area.
>
> **Comparison with Sinc.** We have additionally compared with sinc-activated neural fields [4] on the Kodak image regression task identical to our setup for Table 1; see the table below. Although the official code was not available online, we were able to contact the authors and retrieve the codes (we are very grateful for the help).
>
> We observe that weight-scaled SNFs remain preferable over the baselines.
>
> |  | Xavier Uniform | ReLU+ P.E. | FFN   | SIREN  | Gauss Net | MFN   | WIRE  | NFSL  | Sinc  | Weight Scaling (Ours) |
> |--------|:----------------:|:------------:|:-------:|:--------:|:-----------:|:-------:|:-------:|:-------:|:-------:|:-----------------------:|
> | Kodak | 0.46 | 18.60 | 20.52 | 24.58 | 21.94 | 28.54 | 28.94 | 24.93 | 27.61 | **42.83** |
>
> We will also include the comparison against sinc on other tasks as well, as soon as we finish running the codes.

---

> ### Author Response · Authors · 2024-11-21
>
> ### **W2.** Large-scale experiments
> > **Large Scale Experiments**: The paper compares its methodology of weight scaling on a tasks such as image regression, binary occupancy, spherical data, audio data and fit a nef (in the appendix). Yet I noticed there were no experiments on Neural Radiance Fields (NeRFs) which I found very surprising. The authors say that their method is very effective at reducing the training time of neural fields which is generally large yet many of the tasks such as image regression, audio, occupancy are not considered big tasks within the community. On the other hand NeRF is a big experiment that takes anywhere between 4 hours to 12 hours to train, depending on the fidelity the user wants. I would have liked to see how their method did on such a task that requires hours of training. I also looked at the papers the authors compare to namely [1], [2], [3] and noticed that these papers compared their methods on NeRF. In the appendix the authors do compare their method on "fit a nef" on a large scale dataset but these takes less than 1 hour to train even with a SIREN. Thus I feel another big weakness of the paper is that the authors did not compare their method on a task that takes hours to train such as NeRF, so that we can see how their method does on a more complicated large scale neural field.
>
> Thank you for the thoughtful suggestion. In response to the reviewer’s comment, we have conducted additional experiments on NeRFs and differential equations, which are more large-scale in terms of training time. Briefly, our method demonstrates better performance compared to the baselines.
>
> **Neural radiance fields (NeRFs).**
> We provide the results of the NeRF experiment trained with two different datasets from ‘NeRF-Synthetic’ (as in WIRE). Further details and qualitative results can be found in Appendix F.7.
>
> | Test PSNR after 300 steps  | Lego   | Drums  |
> |----------------|--------|--------|
> | ReLU + P.E.    | 26.02  | 20.97  |
> | FFN		| 26.96| 22.10 |
> | SIREN          | 27.68  | 24.05  |
> | GaussNet       | 25.95  | 22.42  |
> | WIRE           | 26.71  | 23.62  |
> | NFSL           | 27.79  | 24.04  |
> | **Weight Scaling (ours)** | **28.17** | **24.10** |
>
>
> **Solving differential equations.**
> We also performed an experiment on solving wave equations, following the setup of [11]; see Appendix F.8 for the detailed setup. The table below highlights the MSE comparisons (at $t=0$) against the baseline methods. The results (Figure 24) show that weight scaling proves to be effective not only for the given observations but also for previously unseen temporal coordinates.
>
> |       | ReLU + P.E. | SIREN | GaussNet | NFSL | **Weight scaling (ours)**  |
> |-------|:-------------:|:-------:|:----------:|:------:|:-----:|
> | **MSE** |     7.7e-03        |    1.8e-02   |   4.3e-03       |   7.8e-04   |   **2.8e-05** |
>
> ---
> ### **Q1.** The effect of WS on other initializations.
> > Your paper is entirely focused on neural fields that use a sinusoidal activation. The weight scaling method you propose scales the initialization of a sinusoidal neural field by a certain factor $\alpha$, as per eqn (6) in your paper. This method takes the SIREN initialization as its basis. Could you explain what happens if we take a standard initialization scheme like Kaiming uniform or Xavier uniform and scale it in exactly the same way as your eqn. (6)? Does this lead to anything new? Why did you apply your weight scaling to only the SIREN initialization?
>
> We have applied weight scaling on SIREN initialization, as it guarantees favorable theoretical and empirical properties. SIREN initialization theoretically guarantees that the activation distributions remain the same over the layers [11], and this is true even after the weight scaling, as we formalize in Proposition 1. On the other hand, Xavier / Kaiming initializations are designed for linear / ReLU activations, respectively, and thus does not guarantee the same distribution preserving property for sine-activated neural networks. It has also been empirically observed that such initializations perform poorly on SNFs [11,12].
>
> To make this point concrete, we have evaluated the performance (train PSNR) of weight-scaled Xavier / Kaiming-initialized SNFs on the image regression of a Kodak image (see table below); we train for 150 steps, with all other setups identical to the paper. We observe that Xavier / Kaiming initializations lead to a poor performance, with or without weight scaling.
>
> |                | x1   | x2   | x4   | x6   |
> |----------------|-------|-------|-------|-------|
> | Xavier uniform  | 1.29 | 1.29  | 1.30  | 1.29  |
> | Kaiming uniform |-6.88  | -6.91 | -6.91 | -6.92 |
> | SIREN init. |20.69 | 33.18 | 55.19 | 38.55 |

---

> ### Author Response · Authors · 2024-11-21
>
> ### **Q2.** The effect of WS on other activation functions.
> > Could you explain whether your results would go through for other activations that are used within the neural fields literature such as Gaussian [1], Wavelet [2], Sinc [4]?
>
> Thank you for this interesting suggestion.
>
> Theoretically, we do not expect the weight scaling to work satisfactorily with other activations, as we do not have any guarantee that activation distributions will be preserved over the layers. This is in contrast with the sine activations, where WS does not hurt such property, provably (Proposition 1).
>
> Empirically, we have conducted additional experiments and observed that the weight scaling has quite a mixed impact on NFs with other activations. We have tested on a single Kodak image regression task for a single image, as the results are as below.
>
>
> | **Sine (SNF)** | $\alpha=1.0$  | $\alpha=1.5$  | $\alpha=2.0$  | $\alpha=2.5$  |
> |----------|-------|-------|-------|-------|
> | Train PSNR | 20.69 | 24.78 | 33.18 | 54.02 |
> | Test PSNR | 22.15 | 27.67 | 35.57 | 32.39 |
>
>
> |  **ReLU + P.E.**   | $\alpha=1.0$  | $\alpha=1.5$  | $\alpha=2.0$  | $\alpha=2.5$  |
> |----------|-------|-------|-------|-------|
> | Train PSNR | 15.48 | 15.63 | 15.53 | 15.00 |
> | Test PSNR  | 15.95 | 16.10 | 15.98 | 15.34 |
>
>
> | **GaussNet**    | $\alpha=1.0$  | $\alpha=1.5$  | $\alpha=2.0$  | $\alpha=2.5$  |
> |----------|-------|-------|-------|-------|
> | Train PSNR | 22.70 | 27.55 | 28.57 | 27.07 |
> | Test PSNR | 24.88 | 28.62 | 21.97 | 16.76 |
>
>
> |  **WIRE** | $\alpha=1.0$  | $\alpha=1.5$  | $\alpha=2.0$  | $\alpha=2.5$  |
> |----------|-------|-------|-------|-------|
> | Train PSNR | 25.10 | 20.26 | 17.02 | 15.20 |
> | Test PSNR | 14.26 | 11.61 | 10.52 | 9.94 |
>
>
> |   **Sinc NF** | $\alpha=1.0$  | $\alpha=1.5$  | $\alpha=2.0$  | $\alpha=2.5$  |
> |----------|-------|-------|-------|-------|
> | Train PSNR | 24.35 | 29.30 | 29.44 | 29.17 |
> | Test PSNR  | 26.30 | 23.79 | 16.08 | 13.44 |
>
> We observe that weight-scaled GaussNet and Sinc NF also enjoy better training in terms of the training PSNR, but their test PSNR degrades more severely than in SNFs. ReLU+P.E. remains indifferent to WS, and WIRE shows a degraded performance. We have added detailed discussions in the Appendix E.4.
>
>
> ---
> ### **Q3.** Changing frequency as $\omega_h / \alpha$.
> > Your weight scaling essentially just scales the initialization of the inner layers of the initialization by Sitzmann et al for SIRENs. However, if you look at the denominator of eqn. (3), line 194, there is the term $\omega_h$ which is supposed to be the frequency of the sinusoid being used on that layer. Could we not get the same effect as your initialization by using a sinusoid with frequency $\frac{\omega_h}{\alpha}$?
>
> In principle, if we replace $\omega_h$ by $\omega_h/\alpha$ without weight scaling, the initial functional remains the same as SNF initialized with standard SIREN initialization. This is because $\alpha$ terms in the initial weight distribution and the activation function cancel out. In addition, we may expect that the training actually becomes slower; $\omega_h$ is known to serve the role of boosting the weight gradients [11], and thus reducing $\omega_h$ may slow down the training.
>
> To confirm this intuition, we have conducted actual experiments (Appendix E.3). Our empirical results show that such operation makes the training slower than the SIREN-initialized SNF.

---

> ### Author Response · Authors · 2024-11-21
>
> ### **Q4.** Condition number of Hessian
> > In appendix C.3 you analyse the condition number of the Hessian of the loss landscape. You mention that the condition number of the Hessian directly affects the convergence rate. However, my understanding is that this is only true for convex loss landscapes. Loss functions associated to neural networks are almost always non-convex, as they employ non-linearities, therefore in such cases the condition number of the Hessian may not be well defined, as the Hessian may have negative eigenvalues or zero as an eigenvalue. Could you comment on this? I don't understand this analysis you have provided in your paper.
>
> Thank you for pointing this out. On the one hand, the reviewer is correct that the theoretical relationship between the convergence rate and the condition number holds strictly for convex loss landscapes. Thus, the expression that “directly affects” holds for the convex case, but not neural nets; we have fixed this expression.
>
> On the other hand, it is also true that there exists a rich body of empirical and theoretical works that ideas from the convex case, especially regarding the Hessian eigenspectrum, also holds true (or at least insightful) in the optimization of deep neural nets. Precisely, [13] shows that the emergence of outlier eigenvalues when training neural nets; [14] shows that the negative eigenvalues disappear quite quickly during the early phase of training, and outlier eigenvalues slow optimization process in neural network; [15,16] shows that the maximum eigenvalue of Hessian (often referred to as “sharpness”) is indeed closely related to the optimal learning rate, similarly to the convex case; [17,18] study the neural net training techniques (batchnorm and residual connection) through the Hessian, which is quite effective empirically. In this sense, Hessian-based analysis provides a useful perspective toward neural net training. We have clarified this point and the distinction from the “provable results” in the convex case in the revised manuscript.
>
> Furthermore, in Figure 11 of Appendix C.4 (previously C.3), we empirically confirm the observations of [14] on SNFs that the negative eigenvalues of the Hessian quickly disappear after a small number of training steps. Within this context, we have measured the ratio of “absolute” eigenvalues as an approximation of the condition number. While this quantity is not strictly equal to the real condition number, the approximation may help provide useful intuitions to the SNF training.
>
> ---
> [1] Ramasinghe et al., Beyond Periodicity: Towards a unifying framework for activations in coordinate-MLPs, ECCV 2022
>
> [2] Saragadam et al., WIRE: Wavelet implicit neural representations, CVPR 2023.
>
> [3] Saratchandran et al., From Activation to Initialization: Scaling insights for optimizing neural fields, CVPR 2024.
>
> [4] Saratchandran et al., A Sampling Theory Perspective on Activations for Implicit Neural Representations, ICML 2024
>
> [5] Tack et al.,  Learning large-scale neural fields via context pruned meta-learning, NeurIPS 2023.
>
> [6] Andreis et al., Set-based Neural Network Encoding Without Weight Tying, NeurIPS 2024
>
> [7] Thrash et al., MCNC: Manifold Constrained Network Compression, arxiv 2024
>
> [8] Ashkenazi et al., Towards Croppable Implicit Neural Representations, NeurIPS 2024
>
> [9] Ma et al., Implicit Zoo: A Large-Scale Dataset of Neural Implicit Functions for 2D Images and 3D Scenes, NeurIPS 2024
>
> [10] Ji et al., Sine Activated Low-Rank Matrices for Parameter Efficient Learning, arXiv 2024
>
> [11] Sitzmann et al., Implicit neural representations with periodic activation function, NeurIPS 2020
>
> [12] Parascandolo et al., Taming the waves: sine as activation function in deep neural networks, 2016
>
> [13] Sagun et al., Eigenvalues of the Hessian in Deep Learning: Singularity and Beyond, arxiv 2016
>
> [14] Ghorbani et al., An Investigation into Neural Net Optimization via Hessian Eigenvalue Density, ICML 2019
>
> [15] Cohen et al., Gradient Descent on Neural Networks Typically Occurs at the Edge of Stability, ICLR 2021
>
> [16] Zhu et al., Understanding Edge-of-Stability Training Dynamics with a Minimalist Example, ICLR 2023
>
> [17] Lange et al., Batch Normalization Preconditioning for Neural Network Training, JMLR 2022
>
> [18] Li et al., Visualizing the Loss Landscape of Neural Nets, NeurIPS 2018

---

> ### Author Response · Authors · 2024-11-25
> **A Gentle Reminder**
>
> Dear reviewer jBfM,
>
> Thank you once again for taking the time to review our manuscript. We greatly appreciate your time and effort in helping us improve our work.
>
> As there are only 60 hours remaining in the discussion period, we wanted to ask if you have any remaining concerns or questions that we could address. If so, please do not hesitate to let us know.
>
> Best regards,
> Authors.

---

> > ### Comment · Reviewer_jBfM · 2024-11-25
> > **Response by reviewer**
> >
> > Thank you for your response and I apologize for my delayed response. I was actually going through all the updates you made as well as the references you cited in your response to me. I commend you on the extensive experiments you have undertaken however I am still not convinced this is at the level of ICLR. Your paper proposes a simple way to scale the initialization of sine neural fields which while interesting is not enough to be at the level of ICLR.
> >
> > Furthermore, there are points you make to try and prove the novelty of your paper which are simply not true. For example, in your response to me you wrote "In fact, the authors of [1,2,3,4] still utilize sinusoidal activations in their recent work [10]". I went and read the paper [10] and noticed it had nothing to do with using a sinusoid as an activation. It used a sinusoid on the actual weight matrices to increase the rank of the weight matrices. It is NOT using a sinusoid as an activation to activate a layer which is what you are doing within the context of neural fields. Therefore, it is a miss-representation to suggest that [10] is using a sinusoid as an activation. After applying a sinusoid to increase the rank of the weight matrix they then apply another activation for the output of a layer. Thus it is a completely different situation and is rather misleading to suggest it as a piece of new work that applies sinusoids as activations.
> >
> > While the authors have done extensive experiments and have found something interesting I do not believe there is enough for a paper for ICLR. I thank the authors for their conduct and their discussion. I have moved my rating up by one as they have put a lot of effort into producing a variety of experiments.

---

> ### Author Response · Authors · 2024-11-27
>
> Dear reviewer jBfM,
>
>
> Thank you for your continuing efforts to provide useful suggestions to improve our manuscript and also for recognizing our efforts by raising a score.
>
> ***
>
> ### **On not being at the level of ICLR.**
> We strongly believe that our manuscript makes significant and original contributions that are worth sharing with the audience of ICLR. In particular, the key strengths of our paper are:
>
> 1. **Contributes new and significant perspective**: Our work presents a novel perspective that rethinks the role of initialization for speeding up neural field training. Through our exploration, we challenge the common belief that initial weight scales should be strictly determined to a specific value for distribution-preserving properties, to demonstrate for the first time that weight scales can be tuned for a speedup without losing generalization properties. This finding reveals a new and effective design parameter for a better NF initialization, which may inspire much future studies.
>
>
>
> 2. **Theoretical findings and analysis.**: We theoretically show that scaling initialization of SNF does not harm the distribution preserving properties. Moreover, by taking a closer look at weight-scaled functional, we mathematically characterize how the weight scale heavily affects the network properties (i.e., amplifying both relative power of high-order harmonics and early layer gradients) and establish connections to the training speed (i.e., analysis on eNTK).
>
> 3. **Practical impact**: Practically, the proposed weight scaling provides substantial speedup upon the baselines. For instance, on Kodak dataset, training speed is more than 2.9x faster than the best baseline (when measuring #steps until achieving training PSNR 50dB; Fig 25a). As this speedup does not come at the expense of generalization, we strongly believe that our findings bear a significant practical importance.
>
> For these reasons, we sincerely expect that many of the ICLR audience will find our work meaningful and useful.
>
>
> ***
>
>
> ### **On [10].**
> We sincerely apologize for any misunderstanding caused by our response. Our purpose in citing [10] as “in [10], they still utilize sinusoidal activations” was not to emphasize that their proposed network takes the exact form of an SNF, but rather to highlight that their idea shares a similar spirit with that of sinusoidal neural fields: using sine functions as an effective mean to enhance the expressivity.
>
> To elaborate further, the main idea in [10] is to increase the rank of the weights by embedding them within a sinusoidal function with fixed frequency gains. Notably, [10] includes experiments based on neural fields (e.g., NeRF and occupancy field), where sinusoidal functions are applied to the MLP weights, resulting in better performance than a vanilla MLP. This can be interpreted as an attempt to reconstruct the finest details (as well as improving overall reconstruction quality via expanding expressivity) with low-rank matrices, which aligns closely with the practical significance of the sine activation function in SNF.
>
> Therefore, we still believe that sinusoidal neural fields (or at least their core concepts) are widely used in practice, either directly (as mentioned in our previous response) or rather indirectly (as [10]).
>
> ***
> ### **Yet another reason to prefer sine activations.**
> In addition, we note that sine activations can be more preferable for fast training for computational reasons; training with sine activations runs faster on conventional GPUs. This is because their gradients take a simpler form (another sinusoid) than other sophisticated activations, which can be computed fast based on look-up tables (e.g., Volder’s algorithm). To show this point, we have conducted additional experiments to measure the wall-clock time of training a model for 150 steps, on a single NVIDIA A6000 GPU (see below table; averaged over 5 trials). We observe that the sine activations are clearly faster in this aspect. Thus, we still believe that a study on SNF may have meaningful implications, in terms of faster training speed.
>
> | Kodak image regression  | Sine  | GaussNet       | WIRE          | Sinc          |
> |----------------|--------------|----------------|---------------|---------------|
> | **Average training time (sec)**   | **15.20 ± 0.05**   | 20.03 ± 0.07   | 50.85 ± 0.56  | 24.39 ± 0.03  |
>
>
>
>
>
> ***
>
> We sincerely appreciate your active participation during the discussion phase. If there are any remaining questions regarding our response, please let us know.
>
>
> Best,
> Authors.

---

> > ### Comment · Reviewer_jBfM · 2024-12-02
> >
> > I have re-read the authors paper along with the various changes they have made and the various citations they have provided. I do agree that their initialization method is interesting and it does work very effectively and they are right about the citation [10] that it does use a sinusoid function in its architecture. My suggestion to the authors would be to add some of these works to their paper so it comes across to the reader that sinusoidal functions are being used in a variety of areas of machine learning not just in implicit neural representations. This is something I found lacking when reading the introduction and related work. I still feel the contribution is not significant enough though the experimental breadth and and insights do make this an interesting piece of work. I have therefore increased my score by 1. I thank the authors for the ongoing discussion we have had and for taking the time to answer all my questions and making the changes to their work.

---

> > > ### Author Response · Authors · 2024-12-03
> > >
> > > Dear reviewer,
> > >
> > > We deeply appreciate your response, and the feedback regarding including discussions on the use cases of sinusoidal functions in general machine learning outside neural fields. We will include them in the revised manuscript.
> > >
> > > Best regards,
> > > Authors.

---

### Official Review · Reviewer_bvG5 · 2024-10-30

**Soundness:** 3
**Presentation:** 4
**Contribution:** 2
**Rating:** 6
**Confidence:** 2

**Summary:**

The work introduces a method to accelerate training in sinusoidal neural fields (SNFs), focusing on a technique called weight scaling (WS). This method modifies the initialization by scaling the initial weights of SNFs (except for the final layer) by a constant factor. By improving the frequency characteristics of SNFs, WS significantly reduces training time—up to 10 times faster than the traditional approach—while maintaining generalization. The authors provide theoretical insights and empirical validation across tasks in some data domains, including images, 3D shapes, and audio signals. They compare WS with standard initialization and frequency tuning, demonstrating its advantages in both speed and stability during training. The idea behind this method is quite interesting, as it revolves around reducing time to create more efficient models; however, there are some significant weaknesses and gaps that cannot be ignored.

**Strengths:**

* $\textbf{Effective Speed Improvement}$: WS offers a substantial acceleration in training time without degrading model performance or generalization capabilities, which is valuable for SNF applications requiring rapid processing.
* $\textbf{Simplicity and Practicality}$: The method requires minimal changes to existing architectures, making it straightforward to apply WS to SNFs without complex modifications.
* $\textbf{Well-Written}$: The paper has been written in detail.
* $\textbf{Theoretical Justifications}$: The paper offers a detailed theoretical framework explaining why WS is effective, especially in reducing spectral bias and improving the conditioning of the optimization trajectory. This theoretical result supports the empirical findings and strengthens the reliability of the results.

**Weaknesses:**

The authors acknowledge the main limitations of their work, which cannot be ignored. To interpret the results accurately, these limitations must be carefully considered.

**Questions:**

* $\textbf{Impact on Long-Term Generalization}$: How does WS affect long-term generalization on larger datasets or in cases where SNFs are used for predictive tasks (e.g., continuous function approximation or spatial data interpolation)?

**Details Of Ethics Concerns:**

*It seems that the main difference between this work and the work by Sitzmann et al. (2020b) is the weight scaling, specifically the addition of the factor $\alpha$, correct? In the work by Sitzmann et al. (2020b), several results are presented, but this work has only compared a few of them.

*There are some works such as a work by Taheri et al. (2020) which have worked on the scaled neural networks with nonnegative homogeneous activation functions (In particular, ReLU activations).

*In practice, drawing conclusions about a specific structure is challenging for two main reasons. First, conclusions cannot rely on only a few datasets. Second, modifying other aspects of the structure can lead to substantial differences in results. Specifically, changes to a network’s structure&mdash;such as the number of hidden layers, the number of hidden nodes, or the learning rate&mdash;can significantly impact the outcomes.

---

> ### Author Response · Authors · 2024-11-21
>
> Dear Reviewer bvG5,
>
> We thank you for your thoughtful comments and for recognizing the key strengths of our                                                                                           work. Below, we address the concerns you mentioned, point by point.
>
> ---
> ### **Q1.** Impact on long-term generalization
> > **Impact on Long-Term Generalization** : How does WS affect long-term generalization on larger datasets or in cases where SNFs are used for predictive tasks (e.g., continuous function approximation or spatial data interpolation)?
>
> We believe that the reviewer refers to tasks like NeRF (i.e., rendering novel views with interpolated or untrained spatial coordinates) or solving PDEs (i.e., approximating the value of the wave function at unseen temporal coordinates). We have conducted additional experiments on these tasks, which shows that WS indeed improves the performance on these tasks.
>
> **NeRF.** We have evaluated on two data from ‘NeRF-Synthetic’ (as in [1]). Further details and qualitative results have been added to the revised manuscript, in Appendix F.7.
>
> | Test PSNR after 300 steps | Lego   | Drums  |
> |----------------|--------|--------|
> | ReLU + P.E.    | 26.02  |20.97  |
> | FFN		| 26.96| 22.10 |
> | SIREN          | 27.68  | 24.05  |
> | GaussNet       | 25.95  | 22.42  |
> | WIRE           | 26.71  | 23.62  |
> | NFSL           | 27.79  | 24.04  |
> | **Weight Scaling (ours)** | **28.17** | **24.10** |
>
> **PDE.** We have followed the setting of [2] to evaluate the reconstruction quality of the wave equation with partial observations; the table below presents the MSE comparison at $t=0$. To demonstrate the generalization to further time steps, we have also added the quantitative results in the Figure 24 of the revised manuscript, which also confirms the effectiveness of our method.
>
> |       | ReLU + P.E. | SIREN | GaussNet | NFSL | **Weight scaling (ours)**  |
> |-------|:-------------:|:-------:|:----------:|:------:|:-----:|
> | **MSE** |     7.7e-03        |    1.8e-02   |   4.3e-03       |   7.8e-04   |   **2.8e-05** |
>
>
> ---
> ### **Q2.** Additional experiments.
> > It seems that the main difference between this work and the work by Sitzmann et al. (2020b) is the weight scaling, specifically the addition of the factor $\alpha$, correct? In the work by Sitzmann et al. (2020b), several results are presented, but this work has only compared a few of them.
>
> The reviewer is correct that the main difference between our method and the work by Sitzmann et al. (2020b) is how we initialize the SNFs.
>
> Following this suggestion, we have added two experiments that have been missing from our paper and appeared in Sitzmann et al. (2020b): NeRF and PDEs. We have provided experimental results in our response to your previous question, and to the Appendices F.7 and F.8 in the revised manuscript.
>
> Furthermore, we would like to highlight that we also provide two tasks that Sitzmann et al., (2020b) did not cover: Climate data (Table 1), and neural datasets (Appendix F.6). We sincerely believe that this makes our method quite well-validated.
>
> ---
> ### **Q3.** Another work on scaling homogeneous neural networks
> > There are some works such as a work by Taheri et al. (2020) which have worked on the scaled neural networks with nonnegative homogeneous activation functions (In particular, ReLU activations).
>
> We thank the reviewer for the pointer. We have included the reference in the revised version.
>
> ---
> ### **Q4.** About the structure of NN.
> > In practice, drawing conclusions about a specific structure is challenging for two main reasons. First, conclusions cannot rely on only a few datasets. Second, modifying other aspects of the structure can lead to substantial differences in results. Specifically, changes to a network’s structure—such as the number of hidden layers, the number of hidden nodes, or the learning rate—can significantly impact the outcomes.
>
> In fact, this is exactly why we have conducted a variety of ablation studies, ranging over many tasks and modalities (Table 1), resolution (Fig 6(a)), network width (Fig 6(b)), network depths (Fig 6(c)), and the choice of individual datum (Fig 7), to find that scaling up consistently improves the training efficiency of SNF. In the revised version, we have added new ablations on the choice of the learning rates (Appendix E.5) and additional tasks (Appendix F.7 and F.8).
>
> We agree that a careful validation is needed to draw conclusions on a specific structure, and have put our best effort to do so.
>
> ---
> [1] Saragadam et al., WIRE: Wavelet Implicit Neural Representations, CVPR 2023
>
> [2] Sitzmann et al., Implicit neural representations with periodic activation function, NeurIPS 2020

---

> > ### Comment · Reviewer_bvG5 · 2024-11-24
> > **Response to Rebuttal**
> >
> > I thank the authors for adding more experiments to their responses and providing explanations for certain aspects of their work. However, the contribution of this work remains insufficient. My positive grade is based on the results as well as other strengths of the paper, such as its clarity and well-written presentation.

---

> > > ### Author Response · Authors · 2024-11-24
> > >
> > > Dear reviewer bvG5,
> > >
> > > Thank you for the reply, and acknowledging the core strength of our paper: effectiveness, practicality, clarity, and theoretical contributions.
> > >
> > > **Significance of our contributions.** Our paper proposes a new dimension (weight scales) to explore for a better initialization of neural fields. Although the change is quite simple, this dimension has not been well-explored in the existing literature due to a lack of theoretical insight that such an operation does not hurt the signal propagation properties. Our key contribution is the discovery that such an idea can bring massive accelerations in terms of training speed. We sincerely believe that this discovery can inspire many future works on efficiency benefits of neural net initializations.
> > >
> > > Please let us know if you have any remaining concerns, or any further suggestions to strengthen the contributions of our paper. We will be happy to respond, up until the very last minute of the discussion period.
> > >
> > > Best regards,
> > > Authors.

---

> ### Comment · Reviewer_bvG5 · 2024-12-03
> **Response to Rebuttal**
>
> I thank the authors for their efforts. The presentation of the work is now very good, and I have updated the $\textbf{presentation}$ rating to "Excellent".

---

> > ### Author Response · Authors · 2024-12-03
> >
> > Dear reviewer bvG5,
> >
> > We sincerely thank you for your continued effort in reviewing the manuscript and for recognizing the strengths of our work.
> >
> > Best,
> > Authors.

---

### Official Review · Reviewer_svcN · 2024-10-31

**Soundness:** 3
**Presentation:** 2
**Contribution:** 2
**Rating:** 6
**Confidence:** 2

**Summary:**

This paper studies initialization schemes for the class of sinusoidal neural fields (SNFs) and proposes a simple weight scaling method to accelerate training across different data domains. The idea is to simply scale the initial weights of all layers in the neural net by a well chosen factor. Some theoretical results and experimental results are provided to support and demonstrate the proposed method.

**Strengths:**

- Overall the paper is well written, with sufficient number of plots provided to visualize the findings.
- The proposed weight scaling scheme is simple yet seems to be quite effective in speeding up training.

**Weaknesses:**

- The theoretical results are quite limited. For instance, to show that the proposed weight scaling increases the relative power
of the higher-frequency bases, the paper considers the effects of such scaling on a 3-layer SNF only for the case when the width is one.
- While the optimization trajectories are studied via the lens of the empirical neural tangent kernel, it would be more convincing if there are theoretical/empirical results that quantify the impact of such scaling on the gradient loss and the speed of convergence of gradient descent.
- There are also some missing details and confusing parts in the paper that would need to be addressed in order to make the presentation stronger (see my questions below).

**Questions:**

- In Figure 2(b), the relationships between test PSNR and acceleration plotted for both cases of frequency tuning and weight scaling do not seem to be one-to-one. Why is that so? Am I missing something there?
- In Eq. (3), the initialization scheme involves scaling the uniform distributions. Will similar weight scaling trick achieve similar acceleration results when the distributions involved are non-uniform (say Gaussian)?
- How does the speed-up offered by the proposed weight scaling scheme depend on/interact with the choice of optimizers (say SGD vs Adam)? Is it more effective for certain choices of optimizers?
- For the results in Figure 3, the learning rates vary with layers and are scaled down to match the rates of unscaled SNF. Can you provide more details on how they vary with depth? Also, what if the same learning rate is used for all layers?
- Can you provide more details on how the optimization problem (10) is solved? Can we treat $\alpha$ as a trainable parameter instead?

---

> ### Author Response · Authors · 2024-11-21
>
> Dear Reviewer svcN,
>
> We greatly value detailed feedback and thoughtful questions, which have helped clarify and enhance the understanding of our work. We provide detailed responses to the questions below.
>
> ---
> ### **W1.** Theoretical contributions of our work.
> > The theoretical results are quite limited. For instance, to show that the proposed weight scaling increases the relative power of the higher-frequency bases, the paper considers the effects of such scaling on a 3-layer SNF only for the case when the width is one.
>
> We consider a simplified setup because such simplifications lead us to theoretical claims with clear and concise insight. In fact, it is quite common to introduce dramatic simplifications–-as analyzing a neural network with more than two layers remains quite challenging–such as:
> - *Infinite width:* As in [1,2,3]
> - *Two-layer nets:* As in [4,5]
> - *Linear activations:* As in [6,7]
> - *Infinitesimal learning rates:* As in [8,9]
>
> Through such simplification, our lemma 2 & 3 provides a clear yet fresh perspective on how weight scaling affects the initial spectrum of the SNFs. Furthermore, we would like to highlight that our theoretical results are quite well-aligned with the empirical phenomenon, which is rarely true for many theoretical works with complicated settings.
>
>
> ---
> ### **W2.** Discussion about eNTK.
> > While the optimization trajectories are studied via the lens of the empirical neural tangent kernel, it would be more convincing if there are theoretical/empirical results that quantify the impact of such scaling on the gradient loss and the speed of convergence of gradient descent.
>
> We deeply appreciate this suggestion. Following the comment, we have added additional theoretical analysis on the quantitative nature of the eNTK condition number; see Appendix C.3 of the revised manuscript. In this new appendix section, we analyze the eNTK of two-layer SNF to show that the weight scaling reduces the decay rate of the eNTK eigenvalues by characterizing the scale-dependency of the eNTK eigenvalues. This is well-aligned with the empirical observations we provided that the condition number of eNTK is small (Figure 5 and Appendix C.2). The condition number of eNTK can work as a proxy for convergence speed of the gradient-based training in the kernel regime, as described in [10,11].
>
>
> ---
> ### **Q1.** About Figure 2.(b)
> > In Figure 2(b), the relationships between test PSNR and acceleration plotted for both cases of frequency tuning and weight scaling do not seem to be one-to-one. Why is that so? Am I missing something there?
>
> The reviewer is correct that there is no one-to-one correspondence between individual data points for two plots in Figure 2(b). Essentially, this is because our focus here is to illustrate how tuning $\omega$ and $\alpha$ allows us to trade the training speed for generalization; we have chosen the ranges of $\omega$ and $\alpha$ that can help us best capture the full tradeoff curve, which turns out to be not exactly one-to-one. Indeed, we needed to search for a much wider range for frequency tuning ($\omega \in [10,6510]$) than for weight scaling ($\alpha \in [0.4,4.0]$). We have clarified this point further in the revised manuscript, and added further details in Appendix F.1.
>
> ---
> ### **Q2.** The effect of Gaussian initialization.
> > In Eq. (3), the initialization scheme involves scaling the uniform distributions. Will similar weight scaling trick achieve similar acceleration results when the distributions involved are non-uniform (say Gaussian)?
>
> Thank you for the intriguing suggestion. The short answer is: yes.
>
> We first note that, unfortunately, the “Gaussian” version of the principled SIREN initialization does not exist, due to the difficulty in theoretically proving the distribution preserving property theoretically with Gaussian distributions. However, we can conjure up a naïve Gaussian version by replacing the uniform distributions with the Gaussian distribution with the same variance.
>
> We have experimentally tested this variant on a single Kodak image regression task (see the table below). By scaling up the Gaussian-initialized weights, we empirically observe that WS has a similar effect of increasing the train PSNR, and increasing-then-decreasing the test PSNR.
>
> |        | Test PSNR | Train PSNR |
> |--------|:--------------:|:--------------:|
> | Gaussian ($\alpha=1.0$) | 26.53  | 23.91 |
> | Gaussian ($\alpha=1.5$) | 34.13  | 39.79 |
> | Gaussian ($\alpha=2.0$) | 16.55  | 76.34 |
> | Gaussian ($\alpha=2.5$) | 9.35  | 51.48 |

---

> ### Author Response · Authors · 2024-11-21
>
> ### **Q3.** Training with SGD optimizer.
> > How does the speed-up offered by the proposed weight scaling scheme depend on/interact with the choice of optimizers (say SGD vs Adam)? Is it more effective for certain choices of optimizers?
>
> We first clarify that we have used Adam because it trains faster in general, and thus fits our goal of fast training.
>
> With SGD, we confirm that weight scaling remains quite effective. The table below presents the image regression results on Kodak, when trained with SGD (the same setting as in Table 1). The weight scaling clearly outperforms the baselines with SGD, but achieves slightly lower performance than with Adam. We note that WIRE exhibits low performance with the SGD optimizer, which may be addressed through further extensive hyperparameter tuning.
>
> |               | Averaged PSNR |
> |---------------|:---------------:|
> | **SIREN init.**  | 25.10 dB      |
> | **WIRE**  | 14.33 dB      |
> | **Weight Scaling** | 28.42 dB      |
>
>
> ---
> ### **Q4.** About Figure 3.
> > For the results in Figure 3, the learning rates vary with layers and are scaled down to match the rates of unscaled SNF. Can you provide more details on how they vary with depth? Also, what if the same learning rate is used for all layers?
>
> For the $k$th layer of an $l$-layer SNF, we have used the learning rate $\eta \cdot \alpha^{l-k}$. This is inspired by our analytic derivations at the end of Section 4.1 (and Appendix B.3). This scaling down makes the effective learning rate of each layer to be similar, whose performance we visualize in Figure 3(b).
> If we use the same nominal learning rate (i.e., $\eta$) for all layers, then this case is identical to the proposed weight scaling method (Figure 3(a)).
> We have made this point clearer in Appendix F.2. of the revised manuscript.
>
> ---
> ### **Q5.** Details on Eq.(10) and trainable $\alpha$.
> > Can you provide more details on how the optimization problem (10) is solved? Can we treat $\alpha$ as a trainable parameter instead?
>
> In this work, we have explored solving the optimization Eq.(10) by a simple grid search. That is, we have trained weight-scaled SNF with many different values of $\alpha$ and selected the value which maximizes the training speed and meets the constraint. We have tried all $\alpha$ within the range $[1.0,4.0]$ with the grid size 0.2. We have added the details in the  Appendix F.3. of the revised manuscript.
>
> **Considering trainable alpha.** Thank you for an interesting suggestion. We believe that using trainable $\alpha$ to solve Eq.(10) will be quite challenging for two reasons. (1) Even if $\alpha$ is trainable, the function $\mathrm{speed}(\cdot)$ is not differentiable, forcing us to use RL-driven algorithms, as in NAS. (2) Empirically, we observe that training SNF with differentiable $\alpha$ is quite unstable and leads to a poor performance; we have added Appendix E.2 with explicit experiments.
>
>
> ---
> [1] Jacot et al., Neural Tangent Kernel: Convergence and Generalization in Neural Networks
> , NeurIPS 2018
>
> [2] Lee et al., Deep neural networks as gaussian processes, ICLR 2018
>
> [3] Lee et al., Wide Neural Networks of Any Depth Evolve as Linear Models Under Gradient Descent, NeurIPS 2019
>
> [4] Atanasov et al., Neural Networks as Kernel Learners: The Silent Alignment Effect, ICLR 2022
>
> [5] Kunin et al., Get rich quick: exact solutions reveal how unbalanced initializations promote rapid feature learning, NeurIPS 2024
>
> [6] Arora et al., A Convergence Analysis of Gradient Descent for Deep Linear Neural Networks, ICLR 2019.
>
> [7] Min et al., On the Convergence of Gradient Flow on Multi-layer Linear Models, ICML 2023
>
> [8] Chizat et al., On Lazy Training in Differentiable Programming, NeurIPS 2019
>
> [9] Chizat et al., Implicit Bias of Gradient Descent for Wide Two-layer Neural Networks Trained with the Logistic Loss, COLT 2020
>
> [10] Xiao et al., Disentangling trainability and generalization in deep neural network, ICML 2020
>
> [11] Liu & Hui. Relu soothes the ntk condition number and accelerates optimization for wide neural networks. arXiv, 2023
>
> [12] Sitzmann et al., Implicit Neural Representations with Periodic Activation Functions, NeurIPS 2020

---

> ### Author Response · Authors · 2024-11-25
> **A Gentle Reminder**
>
> Dear reviewer svCN,
>
> Thank you once again for taking the time to review our manuscript. We greatly appreciate your time and effort in helping us improve our work.
>
> As there are only 60 hours remaining in the discussion period, we wanted to ask if you have any remaining concerns or questions that we could address. If so, please do not hesitate to let us know.
>
> Best regards,
> Authors.

---

> > ### Comment · Reviewer_svcN · 2024-11-26
> > **Thank you for the rebuttal**
> >
> > I thank the author(s) for providing detailed responses that have mostly addressed my concerns. I have just raised my score.

---

> > > ### Author Response · Authors · 2024-11-27
> > >
> > > Dear reviewer svcN,
> > >
> > > We are glad that our response addressed your concerns, and we sincerely thank you for taking the time to review our work and recognize its value.
> > >
> > > If you have any remaining questions or concerns, please do not hesitate to let us know.
> > >
> > >
> > > Best,
> > > Authors.

---

### Official Review · Reviewer_3nXe · 2024-11-04

**Soundness:** 2
**Presentation:** 3
**Contribution:** 2
**Rating:** 6
**Confidence:** 4

**Summary:**

This work proposes an alternate initialization scheme for sinusoidal neural fields. The main claim is that scaling the initialization weights by a constant value greater than one can help speed up the learning in SNFs as well as improve fitting for higher frequencies, by improving the function at initialization as well as boosting optimization trajectories. Empirical validation on various domains is presented to substantiate the arguments.

**Strengths:**

- The motivation is clear, and the exposition is easy to follow.
- Although not very surprising, the study of the theoretical behavior of weight scaling is fairly comprehensive and I appreciate the authors efforts to draw upon different theoretical tools in existing literature to provide a broad picture.

**Weaknesses:**

- The current choice of experiments is rather limited. Since the contribution is largely methodological, experiments on important practical applications of SNFs, specifically NeRF and solving differential equations (for instance, see Saratchandran et al., 2024), as well as larger datasets, are needed to justify the claims.
- Experimental methodology: A few aspects of the experimental methodology are unclear and/or need revision.
    - Baseline: The included baseline with Xavier initialization is largely irrelevant from a practical standpoint. Alternatively, Multiplicative Filter Networks (Fathony et al, 2021) is an important baseline to consider, especially since multiplicative networks, akin to weight scaling, are known to speed up learning for higher frequencies [4] and thus, directly competes with weight scaling in terms of spectral bias.
    - Fixed Iterations: The choice of number of training iterations to report performance is arbitrary and could reflect unfairly on the baselines. The authors should consider including the PSNR curves for training until convergence (for both WS and baselines) for understanding practical feasibility, since ground truth data to stop at a target PSNR is not available in practice.
    - Full Batch Training/Susceptibility to SGD Noise: In practice, it is often not possible to perform full batch training due to size of dataset, network size etc. How does the noise in moving to stochastic (batched) gradient descent affect the convergence of weight scaling? Experiments that reflect those settings will help understand better the limitations of the proposed framework.
- Related work: Missing citations for Multiplicative Filter Networks (Fathony et al, 2021) as well as spectral bias literature:
    - [2] The Convergence Rate of Neural Networks for Learned Functions of Different Frequencies, Basri et al., NeurIPS 2019
    - [3] Towards Understanding the Spectral Bias of Deep Learning, Cao et al., 2020
    - [4] The Spectral Bias of Polynomial Neural Networks, Choraria et al., ICLR 2022

**Questions:**

- See weaknesses

---

> ### Author Response · Authors · 2024-11-21
>
> Dear Reviewer 3nXe,
>
>
> We sincerely appreciate your valuable feedback and insightful questions, which have enriched our experiments and baselines. Below, we address the questions about our work in detail.
>
> ---
> ### **W1.** Additional experiments.
> > The current choice of experiments is rather limited. Since the contribution is largely methodological, experiments on important practical applications of SNFs, specifically NeRF and solving differential equations (for instance, see Saratchandran et al., 2024), as well as larger datasets, are needed to justify the claims.
>
> Thank you for the suggestion. Following this comment, we have conducted additional experiments on NeRFs and partial differential equations (PDEs). The results confirm that our method shows improved performance over the baselines on these tasks as well.
>
> **NeRF.** We have evaluated on two data from ‘NeRF-Synthetic’ (as in [1]). Further details and qualitative results have been added to the revised manuscript, in Appendix F.7.
>
> | Test PSNR after 300 steps | Lego   | Drums  |
> |----------------|--------|--------|
> | ReLU + P.E.    | 26.02  |20.97  |
> | FFN		| 26.96| 22.10 |
> | SIREN          | 27.68  | 24.05  |
> | GaussNet       | 25.95  | 22.42  |
> | WIRE           | 26.71  | 23.62  |
> | NFSL           | 27.79  | 24.04  |
> | **Weight Scaling (ours)** | **28.17** | **24.10** |
>
> **PDE.** We have followed the setting of [1] to evaluate the reconstruction quality of the wave equation with partial observations; the table below presents the MSE comparison at $t=0$. To demonstrate the generalization to further time steps, we have also added the quantitative results in the Figure 24 of the revised manuscript, which also confirms the effectiveness of our method.
>
> |       | ReLU + P.E. | SIREN | GaussNet | NFSL | **Weight scaling (ours)** |
> |-------|:-------------:|:-------:|:----------:|:------:|:-----:|
> | **MSE** | 7.7e-03 | 1.8e-02 | 4.3e-03 | 7.8e-04 | **2.8e-05** |
>
> Please let us know if there is another dataset which we should additionally evaluate on–we would be more than happy to prepare them, as long as our GPU resources can handle them within the limited time window.
>
> ---
> ### **W2.** The reason for choosing Xavier initialization as a baseline
> > Baseline: The included baseline with Xavier initialization is largely irrelevant from a practical standpoint.
>
> We have included the Xavier init. baseline to highlight the importance of preserving the activation distributions across layers. As [2] points out, classical initializations (e.g., Xavier) fail to retain such properties on SNFs and have poor performance. On the other hand, our initialization preserves the activation distributions, as our Proposition 1 shows. We have clarified this point further in the revised manuscript (Appendix F.4).
>
> ---
> ### **W3.** Using MFN as a baseline
> > Baseline: Alternatively, Multiplicative Filter Networks (Fathony et al, 2021) is an important baseline to consider, especially since multiplicative networks, akin to weight scaling, are known to speed up learning for higher frequencies [4] and thus, directly competes with weight scaling in terms of spectral bias.
>
> We highly appreciate this suggestion. Following this comment, we have additionally compared with MFN (see the revised Figure 1 and Table 1). We find that while MFN achieves impressive performance, our proposed approach still achieves better results. In particular, the revised Table 1 is now as follows:
>
> | Activation         | KODAK (PSNR)   | DIV2K (PSNR) | Lucy (IoU) | ERA5 (PSNR) | Bach (PSNR) |
> |--------------------|----------------|----------------|-----------------|---------------|---------------|
> | **Xavier Uniform** | 0.46±0.10     | 0.39±0.10      | 0.0000±0.0000   | 4.11±0.66     | 7.77±0.20     |
> | **ReLU + P.E.**    | 18.60±0.08    | 16.72±0.08     | 0.9896±0.0003   | 33.30±0.54    | 24.98±0.19    |
> | **FFN**            | 20.52±0.60    | 19.81±0.48     | 0.9843±0.0020   | 38.69±0.27    | 16.66±0.28    |
> | **SIREN init.**    | 24.58±0.05    | 22.86±0.06     | 0.9925±0.0001   | 38.72±0.07    | 37.37±3.11    |
> | **GaussNet**       | 21.94±2.48    | 19.22±0.14     | 0.9914±0.0005   | 38.56±0.51    | 27.47±2.10    |
> | **MFN**            | 28.54±0.10    | 26.42±0.10     | 0.9847±0.0003   | 36.89±0.80    | 16.16±0.05    |
> | **WIRE**           | 28.94±0.21    | 28.20±0.13     | 0.9912±0.0003   | 31.27±0.53    | 16.83±1.85    |
> | **NFSL**           | 24.93±0.07    | 23.39±0.09     | 0.9925±0.0001   | 38.92±0.07    | 37.17±2.88    |
> | **Weight scaling (ours)** | **42.83±0.35** | **42.03±0.41** | **0.9941±0.0002** | **45.28±0.03** | **45.04±3.23** |
>
> We have also added comparisons with the extended number of iterations, for the image regression task; find these at Table 4 in the revised manuscript.

---

> ### Author Response · Authors · 2024-11-21
>
> ### **W4.** About fixed iterations.
> > Fixed Iterations: The choice of number of training iterations to report performance is arbitrary and could reflect unfairly on the baselines. The authors should consider including the PSNR curves for training until convergence (for both WS and baselines) for understanding practical feasibility, since ground truth data to stop at a target PSNR is not available in practice.
>
> Thank you for this valuable suggestion. We have added the full PSNR curve for image regression (for Kodak dataset) to Figure 1 of the revised manuscript, which shows that the proposed method indeed provides a substantial speedup. We are also preparing the full PSNR curve for other tasks, which we are planning to add to the appendix of the revised version.
>
> In addition, we would like to clarify the rationale behind “stopping at fixed iterations.” This criterion is meaningful, as many of the considered tasks are about fitting the seen coordinates (e.g., image regression) where we care about training PSNR. For such tasks, the training PSNR tends to increase slowly but steadily as the training progresses, without a clear convergence point. Thus, for these cases, we find it reasonable to compare the performance after a fixed number of steps. Nevertheless, we agree to the fact that comparing the full PSNR will provide a more informative comparison.
>
>
> ---
> ### **W5.** The effect of mini-batch training.
> > Full Batch Training/Susceptibility to SGD Noise: In practice, it is often not possible to perform full batch training due to size of dataset, network size etc. How does the noise in moving to stochastic (batched) gradient descent affect the convergence of weight scaling? Experiments that reflect those settings will help understand better the limitations of the proposed framework.
>
> **TL;DR.** Our method continues to outperform the baselines with smaller batches.
>
> We first clarify that we have primarily compared full-batch training quite intentionally, as our main focus is the training speed. However, we agree that in many practical cases where we cannot use full batch; indeed, some of our results are already using mini-batches (e.g., occupancy fields).
>
> To provide further insights, we have conducted additional experiments on mini-batch training. The table below summarizes the results across various batch sizes for the image regression task. For smaller batches, we confirm our method continues to outperform the baselines. We observe that weight scaling consistently performs effectively in the mini-batch setting, further demonstrating the robustness of our method. Detailed descriptions, along with PSNR curves, are provided in Appendix E.1.
>
> | Batch Size      | $2^{18}$ (Full batch) | $2^{17}$ | $2^{16}$ |$2^{15}$ | $2^{14}$ | $2^{13}$ |
> |------------------|:-------:|:-------:|:-------:|:-------:|:-------:|:-------:|
> | **SIREN init.**     | 20.69 | 20.19 | 19.53 | 18.99 | 18.48 | 18.03 |
> | **WIRE**     | 25.16 | 22.86 | 20.42 | 16.46 | 14.15 | 13.19 |
> | **Weight Scaling**  | 45.12 | 39.18 | 34.50 | 30.48 | 26.30 | 21.18 |
>
> ---
> ### **W6.** Missing citation.
> > Related work: Missing citations for Multiplicative Filter Networks (Fathony et al, 2021) as well as spectral bias literature.
>
> Thank you for the pointer. We have added the references and discussions to the revised manuscript.
>
> ---
> [1] Saragadam et al., WIRE: Wavelet Implicit Neural Representations, CVPR 2023
>
> [2] Sitzmann et al., Implicit neural representations with periodic activations functions, NeurIPS 2020

---

> > ### Comment · Reviewer_3nXe · 2024-11-22
> > **Response to Rebuttal**
> >
> > - I thank the authors for their efforts and added experiments to address my concerns. However, one of my most pressing concerns still remains in terms of empirical validation outside of the newly added experiments. My primary reason for asking about the PSNR/convergence curves was to understand the limit of performance for these tasks. Specifically, the baselines for both table 1 and table 4 (trained for additional iterations) are massively far off the performance of the proposed method. The current results suggest that all of the other methods can seemingly never achieve the peak performance of WS, which I find hard to swallow and makes me ask whether the implementation of other methods is lacking in some sense (in terms of learning rate, optimizer choice etc.).
> >
> > - This is where having experiments run to saturation (or at least to reasonable extent, which I understand can be subjective) reporting the final PSNR for each baseline helps. Specifically, it demonstrates that while at least some of the baselines are capable of achieving the same peak performance as WS, they require much more compute to do so and makes the speed up story much more concrete. It also helps understand the stability of WS when trained for longer, as a spectral bias towards higher frequencies is known to have worse generalization properties. I do believe it should be feasible for at least some of the considered experiments in the remaining time.

---

> > > ### Author Response · Authors · 2024-11-22
> > >
> > > Thank you very much for the quick reply. We deeply appreciate your efforts to continue providing thoughtful feedbacks.
> > >
> > > We are currently on our way to prepare additional experiments that can resolve your concerns, which we hope to be ready before the end of the response period.
> > >
> > > Before our results get ready, we would like to make several disclaimers:
> > > - *Curves may not saturate easily, for training PSNR.* For many tasks in Table 1 & 4, we are comparing the "training PSNR." Unlike test PSNR (where there are unavoidable generalization gaps) or IoU metrics (which are bounded by 1), it is actually quite common in the literature that the training PSNR curves do not saturate. For example, the Fig 2b, 4a,12 of WIRE reports a curve where the baselines never catch up (https://arxiv.org/pdf/2301.05187). In such regime, fast training directly translates into difficult-to-close gaps. We will try our best to compare the performance at convergence, by (1) trying to train much more extensively, and (2) also comparing the test PSNR, which tend to converge eventually. However, we do want to clarify that this it is not strange if we do not get an easier-to-swallow curves for the training PSNR; if that happens, we sincerely ask for your understanding.
> > > - *Some baselines are not fully reproducible.* During our experiments, we have observed that it is quite difficult to reproduce the reported results for some baselines; some of them did not have officially released codes, and some of them simply did not give the expected results even when we followed the instructions tediously. We will make sure to put addition efforts on hyperparameter tuning, but it is definitely possible that we may fail to make some baselines converge to a satisfactory performance.
> > >
> > > Thank you for your understanding, and we will come back with more results!

---

> > > ### Author Response · Authors · 2024-11-24
> > >
> > > Dear Reviewer 3nXe,
> > >
> > > Thank you for your patience. We now have additional experimental results to resolve your previous comments.
> > >
> > > ***
> > >
> > > ### PSNR curve for training until the performance converges
> > > We report two additional experimental results.
> > > - Image Regression, trained for 5k steps with learning rate schedules. (Appendix F.10 and Figure 25)
> > > - NeRF, with doubled training epochs (Appendix F.7 and Figure 18).
> > >
> > > From the results, we observe that after extensive training—where train PSNR has stabilized and test PSNR fully converged—the proposed WS consistently outperforms the baselines in both training and test PSNR, without collapse in either metric. The performance gap is smaller at convergence, but in terms of training speed (our main goal), the advantage of our method is clear. Also, as our method works well in test PSNR, we believe that the weight scaling with a well-chosen scaling factor $\alpha$ does not hurt the generalization performance (as also demonstrated in Figure 2).
> > >
> > >
> > > **Image Regression.**
> > > We have trained on Kodak image regression dataset for 5k training steps; this is much longer than in recent works, such as GaussNet (3k), WIRE (2.5k), and MFN (1k). Following WIRE, we have adopted the learning rate scheduler, which tends to achieve better performance at convergence (although slightly hurting the early performance). We have also updated the learning rates to the ones that work best with the scheduler.
> > >
> > > From the experimental results (Fig. 25), we make the following observations.
> > > - In Train PSNR (Fig. 25a), the gain from additional training has become very small at 5k steps and the proposed WS continues to outperform the baselines. We note that the PSNRs have not fully saturated. However, as we noted in our previous reply, this is well-expected for training PSNRs in the neural fields literature.
> > > - In Test PSNR (Fig. 25b), we observe that the performance converges at 1k~2k steps. The proposed WS achieves the peak test PSNR very early (37.45dB at 128 steps), and several baselines comparable performance at convergence. However, we note that the baselines require a much larger number of steps to achieve the similar PSNR; FFN requires over 1k steps to achieve the same test PSNR.
> > >
> > > **NeRF.**
> > > For NeRF, we have trained for 600 epochs (i.e., twice as long) using 200$\times$200 (downsampled, as in WIRE) training images. We have carefully tuned the hyperparameters and lr schedules again to account for the changed setup.
> > >
> > > From the results (bottom row of Figure 18), we observe that the test PSNR has been reasonably stabilized at 600 epochs, and the proposed WS consistently outperforms the baselines.
> > > - In Table 5 (below) and Figure 18, we present the quantitative results and test PSNR curve, respectively. In summary, our method consistently outperforms other baselines during extended training, demonstrating its robustness over longer iterations.
> > >
> > > ***
> > >
> > > We sincerely hope that our additional experiments have resolved your concerns. If you have any additional comments, we would be happy to respond further.
> > >
> > > Best,
> > > Authors.

---

> > > > ### Comment · Reviewer_3nXe · 2024-11-25
> > > > **Response to final updates**
> > > >
> > > > I thank the authors for their continued efforts to improve the manuscript. Overall, while the manuscript could still use a fair bit of refinement on the empirical side, the updated subset of experiments does help make the speedup story more palatable. Therefore, I am increasing my score from 5 -> 6.

---

> > > > > ### Author Response · Authors · 2024-11-27
> > > > >
> > > > > Dear Reviewer 3nXe,
> > > > >
> > > > > We are happy to find that our response has addressed your concerns.
> > > > >
> > > > > We sincerely believe that your meaningful comments have greatly helped corroborate our findings.
> > > > >  If you have any further feedback, please do not hesitate to let us know.
> > > > >
> > > > > Best,
> > > > > Authors.

---

### Author Response · Authors · 2024-11-21
**General Response**

Dear reviewers and AC,

We are truly grateful for many insightful and thoughtful comments by the reviewers.

We are also glad that the reviewers found our work to propose a **simple yet effective approach** (svcN, bvG5)  with **solid theoretical foundations** (3nX3, bvG5, jBfM), delivered in  **well-written** (all) manuscript.

***
Through individual responses and revised manuscript, we have carefully addressed the concerns and questions of each reviewer. For your convenience, we provide a short summary of additional experiments, analyses, and clarifications.

**Experiments and Analyses**
- Additional tasks: Our method continues to perform best on these tasks.
    - NeRF.
    - Solving differential equations.
- Additional baselines: Our method outperforms these methods.
    - Multiplicative filter networks.
    - Sinc-based neural fields.
- Additional experimental analyses
    - Ablations on mini-batch
    - Ablations on optimizers: SGD
    - Ablations on initialization schemes: Gaussian / Xavier / Kaiming
    - Ablations on learning rates
    - Analysis on the effect of changing $\omega_h$
    - Analysis on the effect of trainable $\alpha$
    - Applications of WS on other activations: ReLU / Gaussian / Wavelet / Sinc
- Additional theoretical analyses and discussions on:
    - eNTK of Two-layer SNF
    - Eigenspectrum of the Hessian
- Additional visualizations
    - Full PSNR curves

**Clarifications**
- Theoretical contributions of our work. (svcN)
- Experimental details (3nXe, svcN)
- Novelty of our work. (jBfM)
- Hessian-based analyses (jBfM)
- Updates for missing citations. (3nXe, bvG5)

***
In the revised manuscript, following updates are temporarily highlighted in "blue" for your convenience to check. Note that the current version is slightly longer than the page limit; we will compress them after accommodating any remaining request from the reviewers.

We strongly believe that the updates have greatly enhanced the quality of our manuscript.

Sincerely,
Authors

---

### Meta-Review · Area_Chair_S6Fr · 2024-12-10

**Metareview:**

This paper proposes a novel weight scaling initialization scheme for SNFs that speed ups training by scaling the weights of all layers by a constant factor. The authors provide a theoretical analysis and empirical evidence showing that this method improves convergence speed while maintaining or enhancing generalization. The reviewers are overall positive and identified the following strengths and weaknesses:

### **Strenghts**
- The proposed method is straightforward and can be applied without major architectural modifications.
- Demonstrates significant acceleration in training times across various tasks (image regression, 3D shapes, and audio signals) without sacrificing generalization.
- Provides a sound theoretical basis for why weight scaling improves training, including insights into spectral bias and empirical neural tangent kernel analysis.
   - Highlights how weight scaling preserves distributional properties of activations across layers.

### **Weaknesses**
- Work is focused on SNFs and some reviewers argue that SNFs are less commonly used than other neural field architectures in practice.
- Missing or inadequate baselines, such as Multiplicative Filter Networks (MFNs), were highlighted initially. The authors later added this baseline.
- The theoretical results primarily focus on a specific case

### **Reviewer Scores:**
- All reviewers rated the submission marginally above the acceptance threshold, with increased scores after the authors addressed concerns through rebuttals and additional experiments.

The proposed method demonstrates some interesting empirical and theoretical results for accelerating SNFs, but its narrow focus and limited novelty may restrict its impact. The authors addressed many reviewer concerns, including missing experiments and theoretical clarifications, strengthening the manuscript. I therefore recommend acceptance but I highlight there remains some skepticism about its broader applicability and significance, which I encourage the authors to discuss openly in the revised version of the paper.

**Additional Comments On Reviewer Discussion:**

All reviewers rated the submission marginally above the acceptance threshold, with increased scores after the authors addressed concerns through rebuttals and additional experiments.

I believe the authors did a good job at addressing the comments from the reviewers during the discussion period, which brought the paper above the acceptance threshold.

---

### Decision · Program_Chairs · 2025-01-22

Accept (Poster)